# The tumour microenvironment shapes dendritic cell plasticity in a human organotypic melanoma culture

S. Di Blasio [1,11], G. F. van Wigcheren[1,2], A. Becker [1], A. van Duffelen [1,2], M. Gorris [1], K. Verrijp[1,3], I. Stefanini [4,5], G. J. Bakker [6], M. Bloemendal[1,7], A. Halilovic[1,3], A. Vasaturo [1], G. Bakdash[1], S. V. Hato[1], J. H. W. de Wilt [8], J. Schalkwijk[9], I. J. M. de Vries [1], J. C. Textor [1], E. H. van den Bogaard[9], M. Tazzari [1,10,12✉] & C. G. Figdor [1,2,12✉]

The tumour microenvironment (TME) forms a major obstacle in effective cancer treatment and for clinical success of immunotherapy. Conventional co-cultures have shed light onto multiple aspects of cancer immunobiology, but they are limited by the lack of physiological complexity. We develop a human organotypic skin melanoma culture (OMC) that allows real-time study of host-malignant cell interactions within a multicellular tissue architecture. By co-culturing decellularized dermis with keratinocytes, fibroblasts and immune cells in the presence of melanoma cells, we generate a reconstructed TME that closely resembles tumour growth as observed in human lesions and supports cell survival and function. We demonstrate that the OMC is suitable and outperforms conventional 2D co-cultures for the study of TME-imprinting mechanisms. Within the OMC, we observe the tumour-driven conversion of cDC2s into CD14+ DCs, characterized by an immunosuppressive phenotype. The OMC provides a valuable approach to study how a TME affects the immune system.

[1] Department of Tumour Immunology, Radboud Institute for Molecular Life Sciences, Radboud University Medical Center, Nijmegen, The Netherlands. [2] Oncode Institute, Utrecht, The Netherlands. [3] Department of Pathology, Radboud University Medical Center, Nijmegen, The Netherlands. [4] Division of Biomedical Sciences, The University of Warwick, Coventry, UK. [5] Department of Life Sciences and Systems Biology, University of Turin, Turin, Italy. [6] Department of Cell Biology, Radboud Institute for Molecular Life Sciences, Radboud University Medical Center, Nijmegen, The Netherlands. [7] Department of Medical Oncology, Radboud University Medical Center, Nijmegen, The Netherlands. [8] Department of Surgery, Radboud University Medical Center, Nijmegen, The Netherlands. [9] Department of Dermatology, Radboud University Medical Center, Nijmegen, The Netherlands. [10] Immunotherapy–Cell Therapy and Biobank Unit, Istituto Scientifico Romagnolo per lo Studio e la Cura dei Tumori (IRST) IRCCS, Meldola, Italy. [11] Present address: Tumour-Host Interaction Lab, The Francis Crick Institute, London, UK. [12] These authors contributed equally: M. Tazzari, C. G. Figdor. ✉email: carl.figdor@radboudumc.nl; marcella.tazzari@irst.emr.it

Targeting immunomodulatory pathways within the tumour microenvironment (TME) entered centre stage in cancer treatment[1]. Despite promising clinical results of novel cancer immunotherapies, such as immune checkpoint blockade to treat melanoma skin cancer, clinical efficacy is limited and only a minority of patients displays long-lasting clinical responses. It is widely accepted that, in particular, an immunosuppressive TME represents a major hurdle to cancer clearance by immune cells[2–4]. Human melanoma models that resemble the complex tissue architecture can improve our understanding of the contribution of the TME to its immunosuppressed state[5]. This will be pivotal for the design of novel strategies that tackle immunosuppressive networks in melanoma to overcome immunotherapy failure.

Over the past decades, a vast array of experimental approaches has been devised to study the melanoma TME, each presenting unique strengths and flaws[6]. Those models range from two-dimensional (2D) cultures to whole tissue explants. Albeit informative for basic aspects of cancer biology, 2D culture systems are a poor copy of the in vivo cellular environment, as they do not accurately mimic the meshwork of human tissues. Cells in tissue face complex and structurally heterogeneous three-dimensional (3D) architectures, and are exposed to a multitude of cellular and extracellular parameters. Each of those factors can influence tumour growth and the ability of stromal and immune cells to orchestrate immune responses locally[5,7]. Tissue explants obtained from a patient's tumour can be cultured ex vivo for microscopic evaluation, or implanted into immunodeficient mice. Tissue explants and patient-derived xenografts (PDXs) retain cell–cell interactions as well as some tissue architecture of the original tumour and are very useful to monitor natural growth of cancer and to investigate tumour heterogeneity[8]. Nevertheless, PDXs lack a functional immune component, limiting their applicability for the study of therapeutic responses to immunotherapy[9]. On the other hand, tumour explants can only assess pre-existing tumour immune infiltration (as found at the time of tissue resection), and as such only provide a snapshot. In conclusion, the human melanoma microenvironment including its immune cell components is difficult to mimic using current experimental models.

The potential of a human culture system that accurately mimics the in vivo milieu, while having the benefit of a laboratory-controlled environment, has inspired researchers to develop 3D models of skin in which skin cancers, such as melanoma, can be propagated (also referred to as skin equivalents or organotypic cultures). In these models, human epidermal cells and melanoma cell lines, with different invasive capacities, are seeded onto fibroblasts-enriched, animal-derived collagen matrices[10–12]. Others developed organotypic cultures based on acellular, de-epidermized human dermis or self-assembled living sheets made with human fibroblasts secreting their own extracellular matrix (ECM), to obtain a model that more closely resembles the multifaceted skin tissue[13–16]. However, despite important contributions to the field, organotypic skin cultures of melanoma that encompass both stromal- and immunocompetence have never been described.

Within the TME, antigen-presenting dendritic cells (DCs) play an important role in stimulating tumour-specific cytotoxic T cells, thus driving immune responses against cancer[17,18]. Consensus nomenclature for immune myeloid cells classifies human DCs as conventional DCs (cDCs) and plasmacytoid DC (pDC). cDCs can be further subdivided into cDC1 (CD141+ DCs) and cDC2 (CD1c+ DCs) subsets[19]. Despite their crucial role in anti-cancer immunity, evidence suggests that DCs in tumours become largely defective and are no longer capable to alert the immune system to cancer. Furthermore, they are often outnumbered by other myeloid cell subsets, such as tumour-associated macrophages (TAMs) and myeloid-derived suppressor cells (MDSCs) that actively suppress the immune system[20–22]. Moreover, we and others have recently described the enrichment of a myeloid cell population in the circulation of advanced stage cancer patients that co-expresses markers of monocytes/TAMs (such as CD14, CD163) and cDC2s (CD1c)[23–25]. Given their phenotypic distinction from monocytes and macrophages, these cells are called 'CD14+ DCs'[24]. CD14+ DCs infiltrate both primary and metastatic tumour sites, and may attenuate the efficacy of anti-cancer immunotherapies[23]. How these CD14+ DCs are generated remains yet to be determined.

Building on previous knowledge[16], in this study we develop a human organotypic skin melanoma culture (hereafter referred to as OMC), which contains both stromal and immune components. We validate our OMC by exploring functional plasticity of naturally circulating cDC2s that infiltrate the reconstructed melanoma tissue. Interestingly, we observe that the presence of a tumour within the engineered TME leads to the transformation of normal immunostimulatory cDC2s into CD14+ DCs, with a phenotype matching their in vivo counterpart and an impaired ability to stimulate T-cell proliferation. Our results highlight how the generated OMC is instrumental to study tumour-induced events within the TME, that could otherwise not be addressed by the static assessment of tissue biopsies.

## Results

**Human OMC mimics natural primary human melanoma lesions**. A decellularized dermis was used as a scaffold to generate OMCs (Fig. 1a). Histochemical evaluation of dermal markers demonstrated that physical decellularization (outlined in the method section) of the human dermal scaffold did not disrupt its complex extracellular matrix architecture (elastin and collagen fibres), and preserved an intact basement membrane (BM), crucial for adhesion of keratinocytes (KCs) and development of an epidermal layer (Supplementary Fig. 1a, b). Quantitative assessments of the decellularized dermis confirmed the lack of cellular (nuclei) and vascular (CD31) components (Supplementary Fig. 1c–f).

To generate the OMC, primary human fibroblasts (Fbs), KCs and melanoma cells are incorporated onto a decellularized, de-epidermized dermis, and kept under specific culture conditions until a fully differentiated epidermal layer with interspersed tumour clusters is formed (Fig. 1a). To mimic melanoma growth in the OMC, we co-seeded KCs with melanoma cells on the basal membrane layer, at different KC-to-tumour cell ratios. Histological analysis of model tissue sections and comparison to primary human tumour biopsies showed that the KC-to-tumour cell ratio is critical. At low KC-to-tumour cell ratios, extensive proliferation of melanoma cells negatively affects the morphology of the epidermal layer. Co-seeding large amounts of fast-dividing tumour cells with epidermal cells, caused the tumour cells to outnumber KCs, thereby affecting KC differentiation and preventing the formation of a fully differentiated epidermis (Supplementary Fig. 2). The optimal cell seeding concentration, defined as the amount of KCs and melanoma cells that preserves epidermal differentiation and morphology, while allowing the formation of compact nests of tumour cells that infiltrate the underneath dermis, was found to be 25 KCs:1 melanoma cell (Fig. 1b, Supplementary Fig. 2). The proliferation of melanoma cells was determined by a triple Ki67/tumour marker/DAPI staining on OMC sections and Ki67 intracellular staining on the digested OMC suspensions (Fig.1b).

Multiplex fluorescence immunohistochemistry, using two different Fb-associated markers (FSP1 and FAP), along with the tumour marker, showed melanoma cells and Fbs interacting

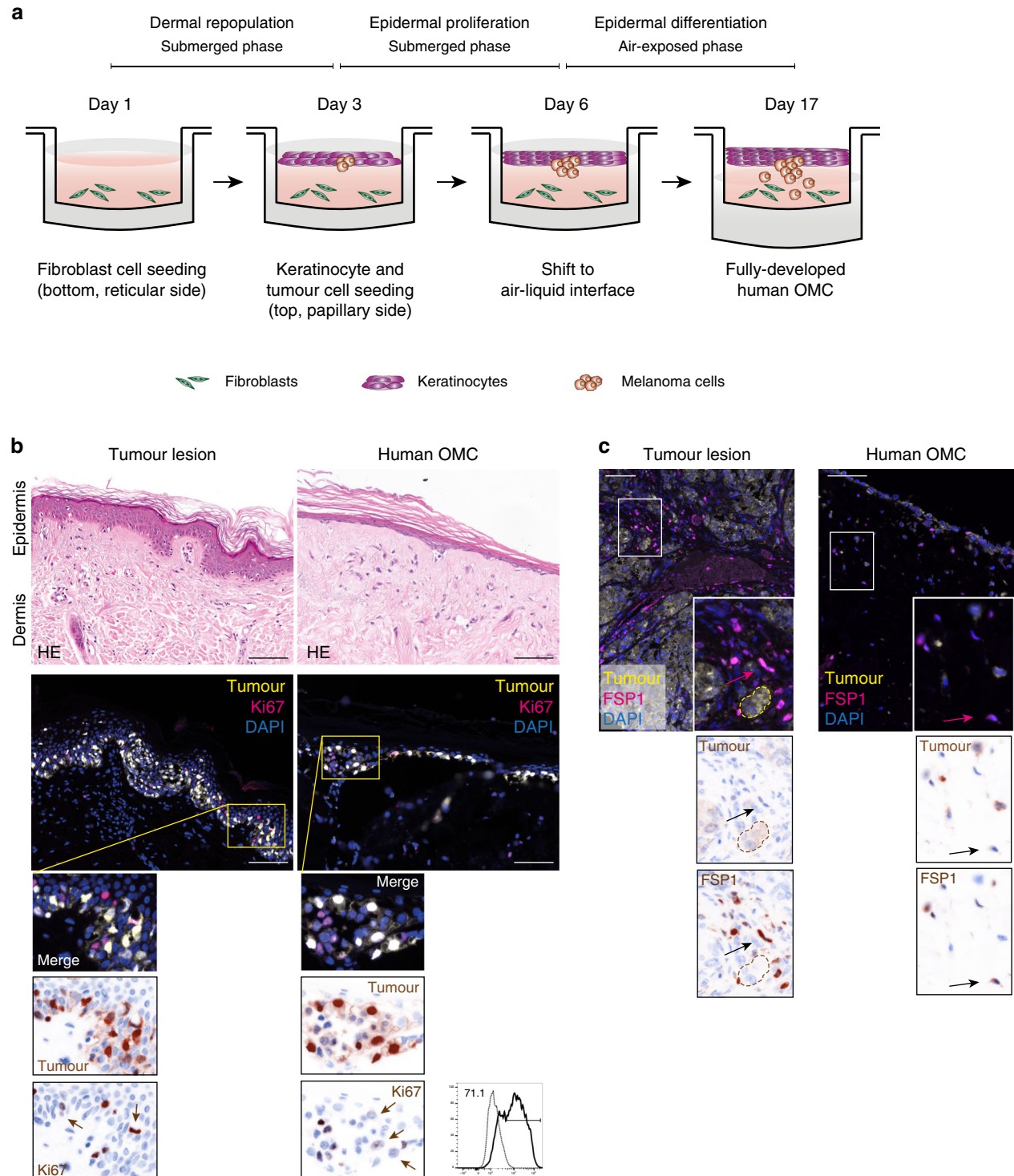

**Fig. 1 Human OMC generation and characterization. a** Overview of the OMC generation (prior to immune cell addition). **b** Histological comparison of tissue sections obtained from primary tumour lesion (left) and OMC (right), showing epidermal and dermal compartments with interspersed tumour cells. Haematoxylin-eosin staining, representative pictures ($n = 3$). Representative areas ($n = 3$) of primary tumour lesion (left) and OMC (right), showing triplex fluorescent staining of Ki67 (magenta), tumour marker (yellow) and DAPI (blue) fluorescence IHC staining. Composite and single colour images of higher magnification are shown. Arrows indicate proliferating tumour cells. Representative flow cytometry histogram of Ki67[+] cells measured within the OMC-digested live CD45-negative fraction. **c** Multiplex fluorescence IHC shows representative area ($n = 3$) of the tumour lesion (left) and the OMC (right) containing fibroblasts (fibroblast-specific protein 1, FSP1[+] cells, magenta) and melanoma cells (tumour marker (tyrosinase and SOX10)[+] cells, yellow). DAPI (blue) indicates nuclei. Inserts show higher magnification. Pseudo-DAB of isolated fluorescent channels of the same areas are shown and fibroblasts are indicated with arrows. Scale bars, 100 µm. Source data are provided as a Source data file.

within the OMC (Fig. 1c, Supplementary Fig. 3). Taken together, these observations show that the OMC mimics the invasive growth of tumour cells into the dermis, closely resembling malignancy-associated lesions in human skin.

**The OMC microenvironment facilitates cell–cell interactions.** A major objective to exploit OMCs is to study human host–tumour interactions. In this study, we incorporated a major subset of DCs directly isolated from the peripheral blood (cDC2s, phenotypically defined as CD1c$^+$CD14$^-$)[23,24], into the dermal compartment of a fully differentiated OMC (day 19, Fig. 2a, Supplementary Movie 1).

After immunohistochemical staining, the distribution of cDC2s was assessed by reconstructing ×20 images into overviews of entire tissue sections (Fig. 2b–d, Supplementary Fig. 4). As a simple metric to quantify the cell density and location within the tissue area (or region of interest, 'ROI'; Supplementary Fig. 4b), we computed the amount of immune cells and other cells (including stromal and epidermal cells), based on cell segmentation and phenotyping using inForm (Fig. 2b, top panel). Enumeration of viable, single cDC2s was obtained with qualitative assessment of signal intensities, analogous to flow cytometry data, for cytometric image analysis as shown in Fig. 2b (bottom panel), followed by microscopic evaluation. Live cells were defined as Cl. Cas 3$^-$, based on Cleaved-Caspase-3 staining, single immune (CD45$^+$ DAPI$^+$) events (Fig. 2c). We found that cDC2s were homogeneously distributed throughout the reconstructed melanoma tissue (±20 cDC2s counted per mm$^2$) (Fig. 2d). Altogether, these findings demonstrate that the OMC is a potentially valuable tool to study immune cells within the TME.

**DCs interact with melanoma cells in a human OMC environment.** Immunohistochemistry end point analysis of fixed OMCs revealed that cDC2s in the dermal compartment were in close proximity with both melanoma cells and Fbs (Fig. 3a). To confirm that cDC2s actively interact with cells in their surrounding niche and engage in cell-to-cell interactions with tumour cells, we exploited live two-photon microscopy with a time-lapse setting (Fig. 3b–f, Supplementary Fig. 5). Combination of fluorescent signal (live-cell visualization) with second harmonic generation (SHG, elicited by collagen fibre bundles) delivers information on the 3D anatomy of OMCs. Like in natural skin, collagen fibres were organized in heterogeneous networks, including randomly arranged loose collagen fibres and aligned in more compacted collagen bundles (Fig. 3b, Supplementary Fig. 1a, Supplementary Fig. 5a, d). cDC2s were highly dynamic, as evidenced by a typical hand-mirror shape (Fig. 3c, Supplementary Fig. 5b, c), characteristic of amoeboid motility. cDC2s displayed a leading edge, consisting of multiple dendrites that intercalated between tissue structures, followed by the cell body containing the nucleus, and a posterior tail (uropod). Analysis of DC shape and degree of cell protrusion did not reveal significant differences with respect to their distance from a tumour (Supplementary Fig. 5a–c). Furthermore, we observed that GFP-expressing melanoma BLM cells (BLM-GFP) showed intense membrane dynamics, evidenced by protrusive and retractile activity (Fig. 3d). Interestingly, cDC2s in our OMC actively interacted with live tumour cells and even sampled tumour-derived cellular microparticles (or blebs), released by the BLM-GFP cells into the ECM (Fig. 3e, f, Supplementary Movie 2,3 and Supplementary Fig. 5d–f). Dynamic DC-tumour cell interactions were observed as early as a few hours after immune cell injection, over periods of at least 30 min (Fig. 3e, Supplementary Fig. 5f). Tumour blebs remained intact as evidenced by the retained cytoplasmic GFP. Taken together, these

observations clearly indicate that the OMC represents a promising and valuable platform for accurate investigation of cellular interplay and function, in an in vivo-like skin tissue architecture.

**cDC2s convert into CD14$^+$ DCs in human OMCs.** We and others have previously described the enrichment of CD14$^+$ DCs in metastatic melanoma, leukaemia and breast cancer patients[23,24,26]. Giving their clinical relevance in different tumour types, we next evaluated whether direct tumour influence in the OMC could convert immunocompetent cDC2s into CD14$^+$ DCs. Nowadays, cDC2 directly isolated from peripheral blood are used to prepare DC vaccines to treat cancer patients[27,28]. Moreover, given their relevance in driving anti-tumour T-cell responses, also in the context of checkpoint inhibitor therapy[29], it would be important to know if these immunostimulatory cells have the potential to become immunosuppressive within the TME.

To test this, we studied how the TME of three different melanoma cell lines (BLM, Mel624, A375) modulated the phenotype and function of immunocompetent cDC2s. Tumour-free organotypic skin cultures (OSCs) were generated as controls. We obtained highly purified cDC2s from healthy donors (purity >96%. Supplementary Fig. 6a), including an additional step of CD14$^+$ cell depletion, and injected them into OSCs or OMCs. After 2 days of culture, organotypic cultures were enzymatically and mechanically digested. Supplementary Fig. 6b shows the gating strategy applied for the discrimination of live immune CD45$^+$ and non-immune CD45$^-$ cells in the digested OSCs and OMCs.

Interestingly, we observed induction of the CD14 monocytic marker on cDC2s isolated from OMCs, compared with cells harvested from OSCs (Fig. 4a). Figure 4a shows representative contour plots identifying two distinct populations: cDC2s (CD1c$^+$ CD14$^-$ cells, orange) as originally injected, and cells converted into CD14$^+$ DCs (CD1c$^+$CD14$^+$ cells, green). Pie charts in Fig. 4b illustrate how, in the presence of melanoma, frequencies (mean ± SD) of cDC2s and CD14$^+$ DCs change, towards an accumulation of the latter and a concomitant prominent reduction of cDC2s in the TME, compared with OSCs. This phenomenon was consistently observed throughout all experiments and independent of the melanoma cell line used (Fig. 4c). Time-course analysis revealed that the accumulation of CD14$^+$ DCs occurred already upon overnight culture in the OMC, and further increased over time (48 and 72 h after injection) (Supplementary Fig. 7a, b). To better define these cells, we assessed whether differences in the expression of the monocytic marker, CD14, reflected additional changes characteristic for CD14$^+$ DCs. We observed that these cells displayed lower CD1c, CD86 and HLA-DR (GeoMFI CD14$^+$ DCs/cDC2s ratio <1.0) and also expressed higher levels of PD-L1 (GeoMFI CD14$^+$ DCs/cDC2s ratio >1.0) (Fig. 4d). Interestingly, besides CD14 upregulation, we found that CD14$^+$ DCs expressed higher levels of markers typically associated to TAMs: CD163, CD206 and MerTK (GeoMFI CD14$^+$ DCs/cDC2s ratio >1.0) (Fig. 4e). Of note, changes in PD-L1 and CD163 expression were not observed as early as 8 h, but became evident at longer exposure time within the TME (Supplementary Fig. 7c).

To determine whether soluble factors might be responsible for cDC2 conversion, we exposed cDC2 DCs to conditioned media (CM) collected from BLM tumour cells and OMC, and observed that both CM drove the conversion of freshly isolated cDC2 into CD14$^+$ DCs (Fig. 5a), as opposed to control medium. We therefore examined secretome profiles of tumour-conditioned (BLM-CM and OMC-CM) and control media (Fb-CM and OSC-CM) (Fig. 5b), and focused our attention on CCL-2, IL-6 and PGE2, factors known to play a role in tumour modulation of myeloid functions[30–32]. Particularly, while CCL-2 was found

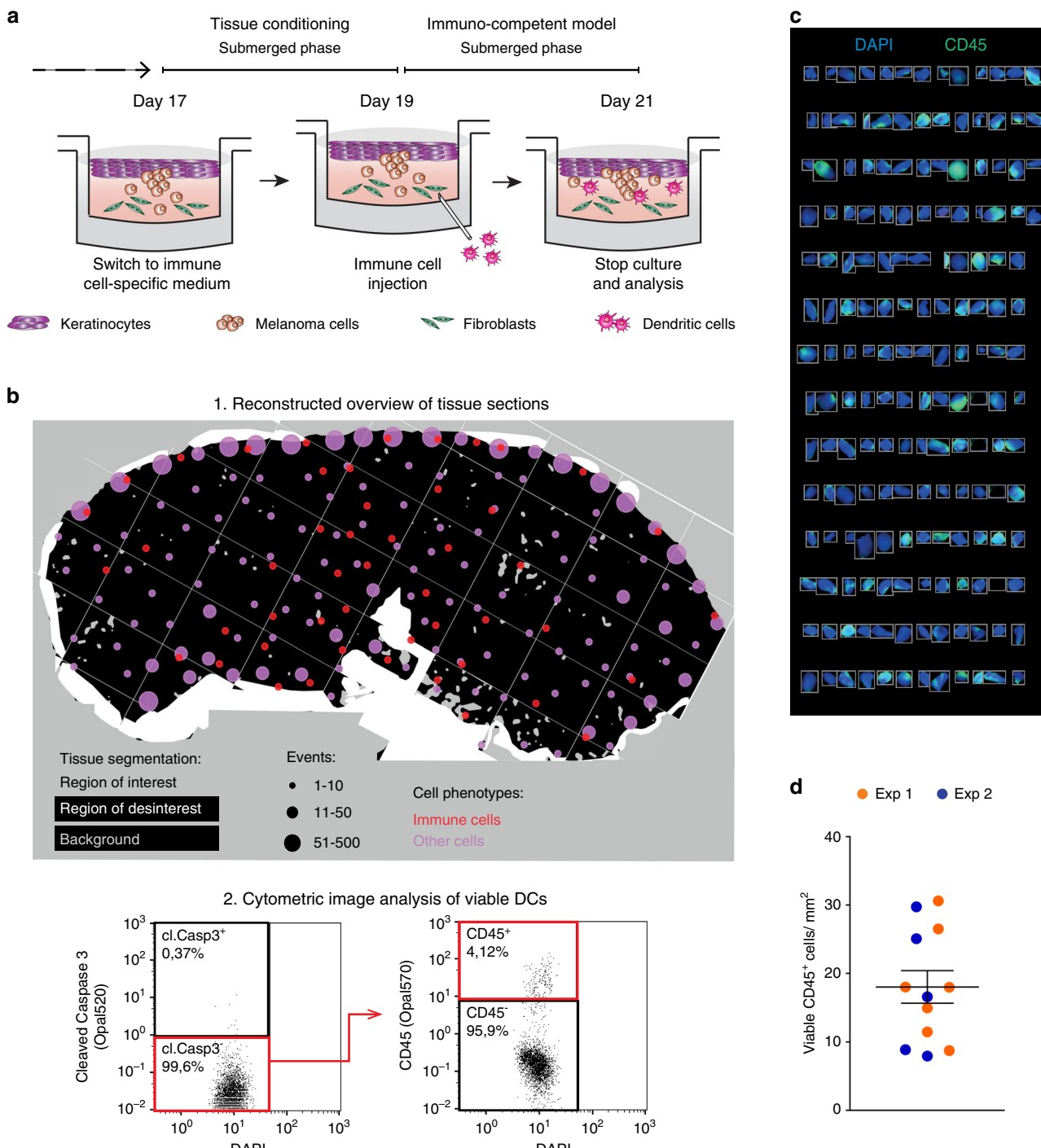

**Fig. 2 The OMC microenvironment sustains immune cell survival and distribution. a** Overview of the experimental approach used to obtain an immunocompetent OMC. **b** Tissue classification and DC localization within the selected ROI. Coloured dots indicate distribution of immune (red) and other cell types (pink), based on tissue segmentation and positivity score (upper panel, 1); cytometric image analysis and gating strategy to identify viable DCs (DAPI+ Cleaved-Caspase-3− CD45+ events) (lower panel, 2). **c** Viable cDC2s as a result of the gating strategy visualized in the original image, showing a representative selection of gated events. DAPI (blue), CD45 (green). **d** Quantification of viable, immune cells/mm² in at least 5 sections per construct, wherein colours indicate two independent experiments. Source data are provided as a Source data file.

co-expressed amongst all CM tested (OMC-CM, BLM-CM, Fb-CM and OSC-CM); IL-6 was mainly detected in OMC-CM and BLM-CM. Moreover, we observed that PGE2 was selectively expressed in OMC-CM, but completely absent in the other conditions. This was in line with the observation that only rhIL-6 and rhPGE2, but not rhCCL-2, drove the conversion of

stimulatory cDC2s towards immunosuppressive CD14+ DCs (Fig. 5c). The role of rhIL-6 in modulating the generation of CD14+ DCs was confirmed by the significant reduction induced upon addition of the IL-6 blocking antibody (Fig. 5d). In addition, IL-6 blockade also partially abrogated the immunosuppressive modulation of cDC2 induced by either BLM-CM or OMC-CM.

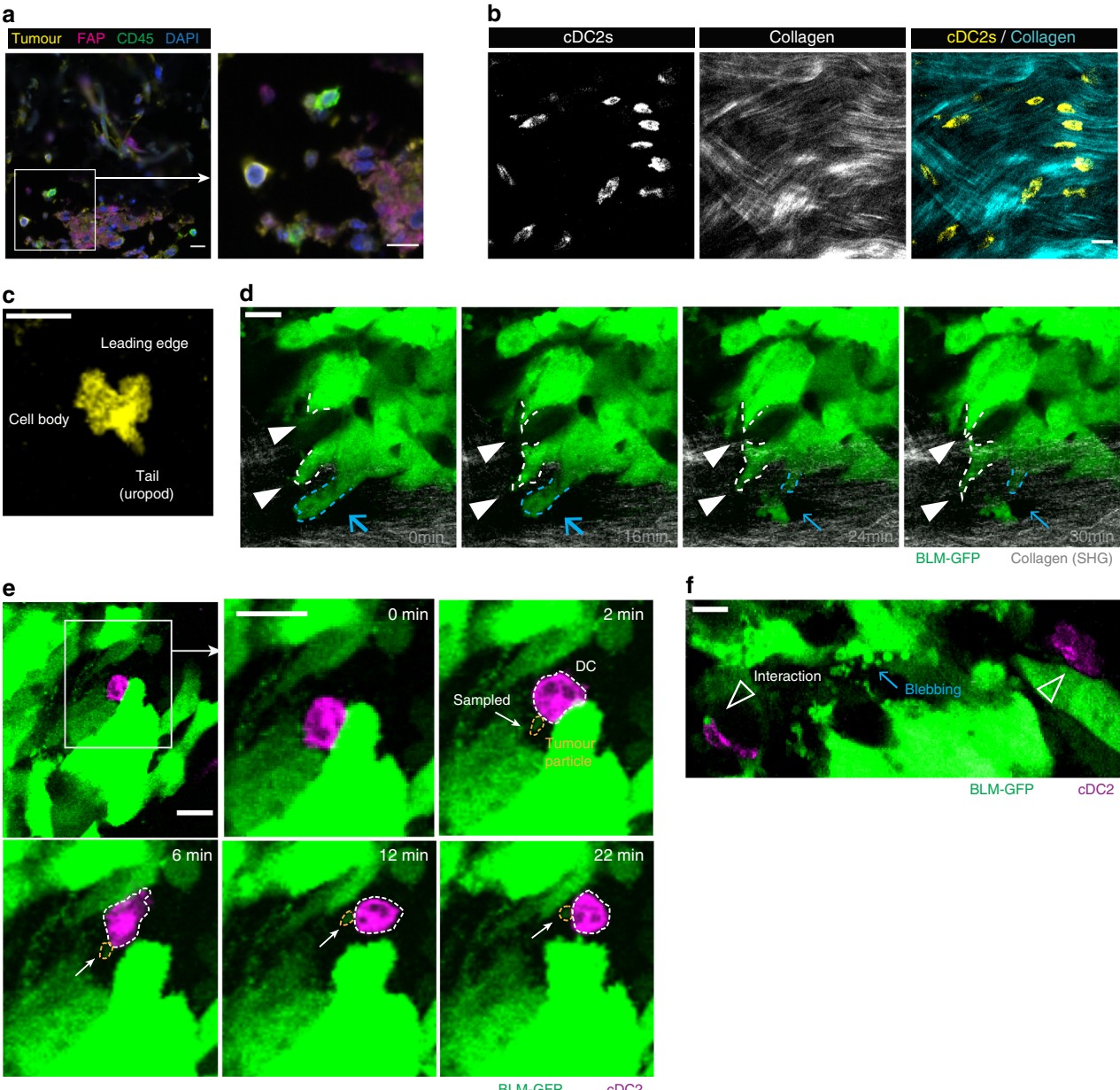

**Fig. 3 Visualization of cDC2-tumour cell interaction. a** Section of OMC showing multiplex fluorescence immunohistochemistry representative area. cDC2s (CD45[+] cells, green) are found in close proximity to melanoma cells (tumour (tyrosinase and SOX10)[+] cells, yellow) and fibroblasts (FAP[+] cells, magenta). DAPI (blue) indicates nuclei ($n = 2$). **b** PKH26 (yellow) and second harmonic generation (SHG, cyan) signals of multiphoton images, acquired with excitation wavelength ($\lambda$) of 950 nm. PKH26 signal indicates cDC2s; SHG was generated by collagen bundles. **c** Characteristic amoeboid behaviour of DCs patrolling the environmental niche. **d** Representative time points during time-lapse recording of melanoma BLM cells expressing GFP [$\lambda$ (excitation) = 950 nm]. BLM cells showed high cellular dynamics (dotted lines; protrusion, white arrowhead and retraction, blue arrow). **e, f** Representative time points during time-lapse recording of cDC2s (PKH26)-tumour cell (GFP) interaction [$\lambda$ (excitation) = 950 nm]. **e** DC (dotted line, white) sensed and sampled tumour-derived particle (dotted line, yellow). **f** Prolonged interaction of DCs with tumour-derived fragments. Tumour cells showed intense membrane dynamics and blebbing. Scale bars, 20 μm.

In order to further investigate the importance of dimensionality in the tissue microenvironment, we performed intra-donor comparisons ($n = 4$) of cDC2s, isolated from OMCs, to those co-cultured with the same tumour cells in conventional 2D co-cultures. Percentages of CD14[+] DCs were significantly lower in 2D co-cultures compared to those isolated from OMCs (Supplementary Fig. 8a, b), providing additional evidence for the importance of 3D organotypic cultures that clearly mimic a tissue microenvironment. Moreover, CD14[+] DCs generated in 2D always failed to upregulate CD163 and MerTK, and had a

much lower expression of PD-L1 compared with cells cultured in the OMC (Fig. 5e and Supplementary Fig. 8c). Altogether, these data demonstrate that (1) within the OMC the presence of a tumour drives cDC2s towards a different myeloid cell subset phenotypically resembling CD14[+] DCs, (2) the observed phenomenon is tumour-dependent since OSCs lacking tumour cells contain significantly less cells expressing CD14, (3) tumour-induced secretion of IL-6 and PGE2 is at least in part responsible for cDC2 DC conversion, (4) the cross-talk between melanoma cells and fibroblasts, within the reconstructed OMC, causes

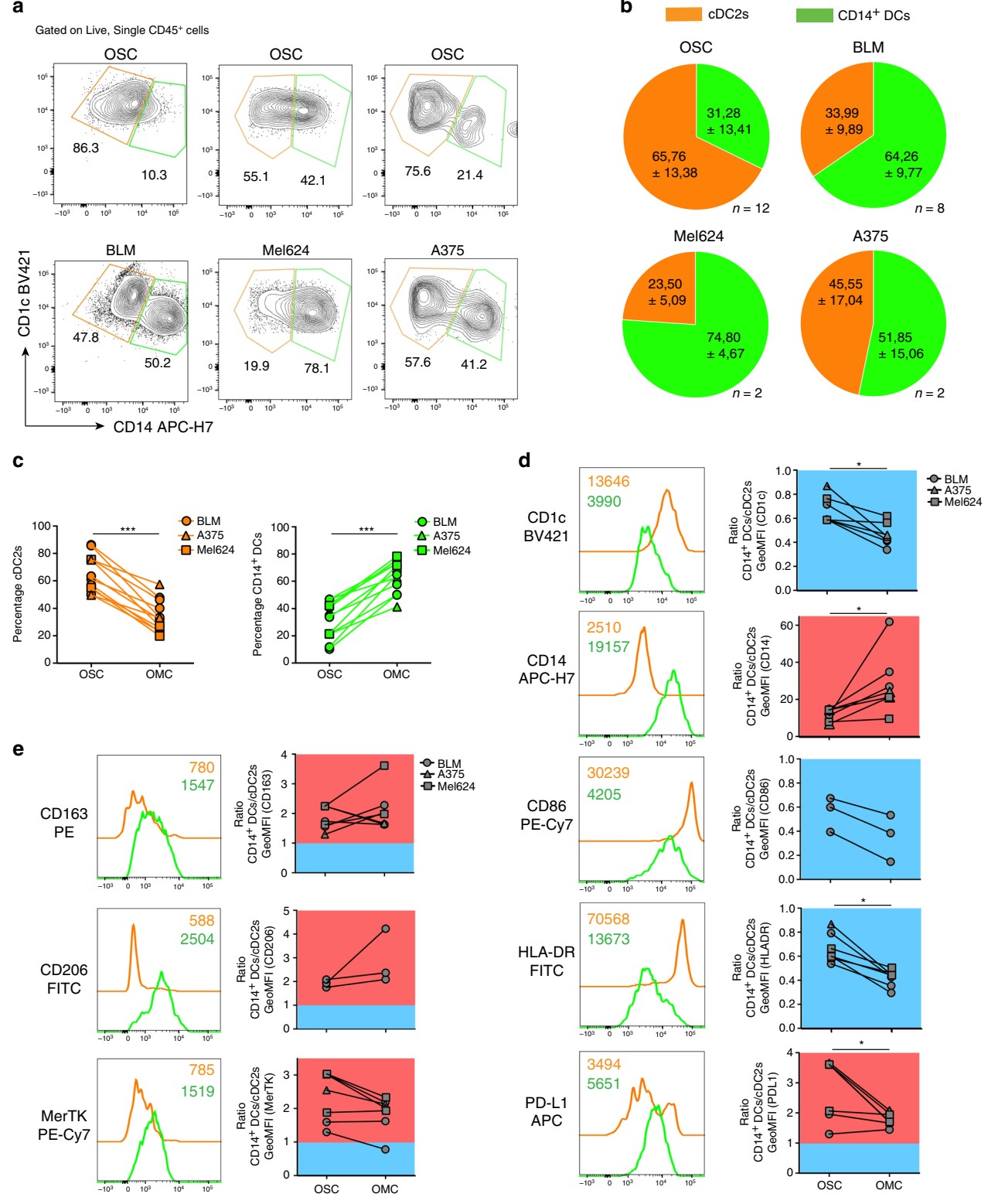

increased secretion of PGE2, underlining the added value of a multicellular microenvironment system and (5) 3D multicellular skin cultures are essential, as conventional 2D co-cultures do not recapitulate the complete CD14+ DC phenotype.

**Tumour-induced CD14+ DCs display immunosuppressive function.** To functionally characterize tumour-induced CD14+ DCs, we next performed qRT-PCR on cDC2s and CD14+ DCs

FAC-sorted 18 h post-injection. We confirmed at the transcriptomic level that CD14+ DCs from OMCs express CD14, CD163 and CD206 genes (Supplementary Fig. 9a), consistent with protein surface expression levels as measured by flow cytometry (Fig. 4c, d). Interestingly, we observed that these CD14+ DCs generated within the OMC also expressed higher levels of markers previously linked to human myeloid cells endowed with suppressive activity: *XBP1, THBS1, IL-6, PTGS2,* and *HIF1A*

**Fig. 4 Identification and phenotypic characterization of CD14+ DCs isolated from the human OMC. a** Two days after cDC2s injection, OMCs were digested and cell suspensions were stained. Representative flow cytometry contour plots showing gated cDC2s (CD1c+CD14− cells) and CD14+ DCs (CD1c+CD14+ cells) in OSCs (upper contour plots) and OMCs (lower counter plots). Results obtained with three different melanoma cell lines (BLM, Mel624, A375) are shown. Numbers indicate the percentage of gated cells. **b** Pie charts depicting the frequency of cDC2s and CD14+ DCs in OSCs and OMCs. Independent experiments were performed with three different melanoma cell lines: BLM ($n = 8$), Mel624 ($n = 2$), A375 ($n = 2$). Frequencies from control OSCs are pooled ($n = 12$). Data are mean ± SD. **c** Graphs showing the frequency of cDC2s ($p = 0.0005$) and CD14+ DCs ($p = 0.0005$) in OSCs and OMCs (across the three melanoma cell lines). A non-parametric two-tailed Wilcoxon signed-rank test was used for statistical analysis. **d** Representative overlaid histogram plots showing expression of the indicated markers in cDC2s and CD14+ DCs isolated from OMCs. For each marker a summary graph reporting the ratio of the geometric mean of the fluorescence intensity (GeoMFI) between CD14+ DCs and cDC2s isolated from OSCs and OMCs is shown. A blue background indicates a GeoMFI ratio <1, higher expression in cDC2s; whereas a red background indicates a GeoMFI ratio >1, higher expression in CD14+ DCs. At least $n = 3$ donors for each marker are included. The $p$ values were estimated using a non-parametric two-tailed Wilcoxon signed-rank test. **e** Representative histogram plots and GeoMFI graphs of macrophage-related markers CD163, CD206 and MerTK in cDC2s and CD14+ DCs isolated from OMCs. At least $n = 3$ donors for each marker are provided. The $p$ values were estimated using a non-parametric two-tailed Wilcoxon signed-rank test. Statistical significance was annotated as follows: *$p < 0.05$, ***$p < 0.001$. Source data are provided as a Source data file.

(Fig. 6a, Supplementary Fig. 9b)[30,33]. At the protein expression level, we confirmed that only CD14+ DCs promptly produced IL-6 (11.1% vs 0.53%), upon TLR4 stimulation (Fig. 6b). Intriguingly, we observed that these cells could also be distinguished based on their S100A9 expression (Fig. 6c), underlining another similarity to already described regulatory myeloid cell features[34]. Of note, the immunosuppressive molecule IDO1, often found in association with tolerogenic DCs[35] was predominantly expressed in cDC2s that still retained a clear DC phenotype upon interaction with tumour cells (Fig. 6a). This suggests that within OMCs, injected cDC2s follow two different fates: (1) they become tolerogenic cDC2s; (2) they convert to CD14+ DCs. To further investigate if these cDC2s also functionally reverted to immunosuppressive myeloid cells, we performed mixed lymphocyte reactions to test their immunostimulatory potential. cDC2s that acquired CD14 during culture, were less capable of stimulating allogeneic CD3+ T cells when compared with cDC2s that lacked CD14 (Fig. 6d), and this stimulatory ability correlated with their HLA-DR expression (Fig. 6e). Similarly, they were significantly less able to induce autologous T-cell activation (Fig. 6f). Of note, we observed that CD14+ DCs, isolated from OSCs, were already less stimulatory than cDC2s, but the presence of the tumour further decreased their ability to stimulate T-cell proliferation and activation (Supplementary Fig. 9c, d). Collectively, our data show that cDC2s that acquire CD14 within OMCs, represent a functional distinct myeloid population induced by the TME.

**CD14+ DCs in OMCs resemble those found in melanoma lesions.** Subsequently, we set out to compare CD14+ DCs induced in the OMC with myeloid cells isolated from melanoma lesions of stage IV metastatic melanoma patients. Using flow cytometry analysis, we identified CD14+ DCs (green) and cDC2s (orange) in melanoma lesions, similarly to what we previously observed within OMCs (Fig. 7a). Pie charts in panel b summarize the results obtained across three different patients. Further phenotypic analyses revealed the macrophage-like nature of these ex vivo CD14+ DCs, characterized by a higher expression of CD163, CD206 and MerTK (Fig. 7c). Histograms in panel c show that ex vivo CD14+ DCs, identified within intra-tumoural CD45+ CD11c+ cells (Supplementary Fig. 10), have a phenotype that is closer to that of CD14+ monocytes/macrophages (mono/macs, red), rather than cDC2s. Moreover, we could confirm that also ex vivo, CD14+ DCs expressed higher levels of PD-L1 when compared to cDC2s. Overall, this ex vivo analysis of tumour biopsies highlights the phenotypic similarity between CD14+ DCs reprogrammed within OMCs and those isolated from melanoma lesions.

**Patient-derived OMCs educate autologous monocytes.** Finally, we explored the suitability of the organotypic culture conditions for different immune and non-immune cell subsets. In this respect, we tested whether the dermal scaffold allows culture of human lymphocytes, isolated from peripheral blood and metastatic melanoma lesions. We observed that peripheral blood CD3+ CD4+ and CD3+CD8+ T cells could be recovered 2 days after injection, with no loss or changes in the proportion of T-cell subpopulations, compared with conventional 2D cultures (Supplementary Fig. 11). Similarly, the frequencies of melanoma-infiltrating lymphocytes, cultured for 2 days in a dermal scaffold (Fig. 8a, b), were in line with the parallel ex vivo phenotypic analysis of patient samples (Supplementary Fig. 12a). Furthermore, we used pairs of prospectively collected tumour material and peripheral blood from melanoma patients ($n = 3$), for the de novo generation of autologous patient-derived organotypic models. In particular, CD45− cells (tumour and stromal populations) were injected in a dermal scaffold, and cultured for 48 h; followed by injection of autologous blood circulating CD14+ monocytes (as a source of myeloid immune cells) and an additional 2day culture period. Analysis of patient-derived models showed that HLA class I, Ki67 and expression of four tumour antigens on melanoma cells was maintained, as compared with the original tumour lesion (Fig. 8c, d and Supplementary Fig. 13). A similar level of tumour-imprinting in infiltrating immune cells was observed in this autologous setting, confirming and extending the observations made in the melanoma cell line-derived OMCs. Blood circulating CD14+ monocytes underwent the same phenotypic conversion as we observe for cDC2 DCs (HLA-DR downregulation, and upregulation of CD14, MerTK and PD-L1) when infiltrating the TME (Fig. 8d and Supplementary Fig. 12b). Altogether, our results demonstrate that patient-derived organotypic models resembled melanoma lesions in terms of tumour-specific characteristics, composition of tumour-infiltrating lymphoid cells, and the ability to induce changes in the phenotype of myeloid cells.

## Discussion
Three-dimensional, organotypic cultures of cancer types have the advantage of recapitulating both histological and mutational features of the original tumour, and can be propagated for extended periods of time, thus facilitating experimenting[36–38]. Yet, they fail to represent the multicellular composition of the TME; in particular, the inclusion of stroma and tumour-infiltrating lymphocytes has been only recently described[39]. Among skin malignancies, melanoma is difficult to mimic due to the structural complexity of the skin tissue. In the context of human melanoma, local tumour growth was achieved in in vitro skin equivalents, which, however, lacked an immune

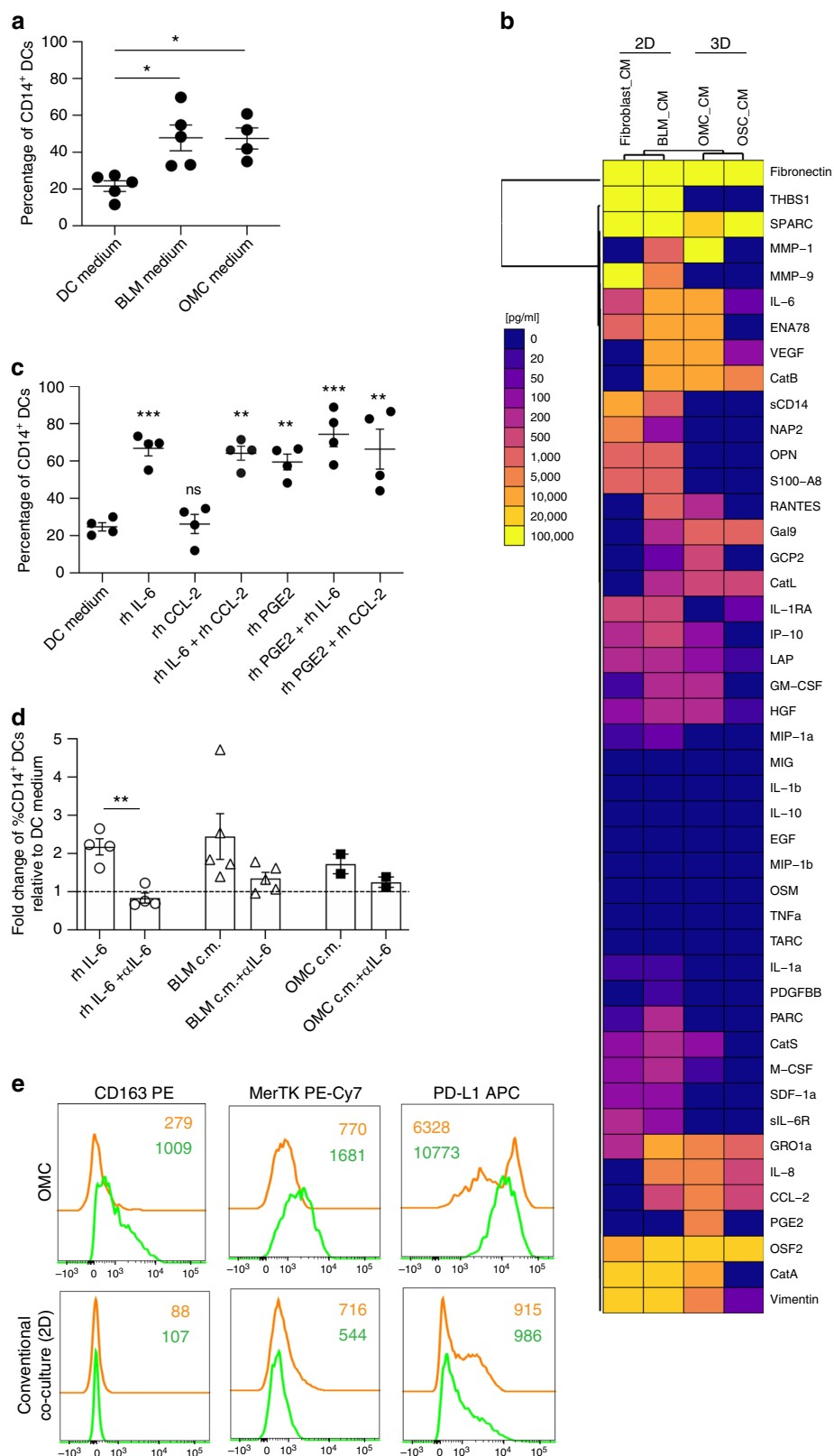

compartment[16]. With the aim of filling this gap, we generated a human multicellular skin microenvironment, amenable to controlled experimental manipulation, suitable to explore cancer-driven immune cell modulation within a complex tissue architecture. Similar to organotypic tumour cultures, the OMC we developed, reproduces the spatial distribution of cells that naturally constitute the skin. We observed that an optimal KC-to-tumour cell ratio is essential to ensure the formation of the epidermal layer, while allowing proliferation of tumour cells into nests that infiltrate the underneath dermis. The decellularized human tissue employed here provides natural skin stiffness and tissue architecture of collagen and elastin fibres, advancing the

**Fig. 5 Effect of soluble factors and dimensionality on the CD14$^+$ DC conversion and phenotype. a** Frequency of CD14$^+$ DCs upon culture of cDC2s in DC medium ($n = 5$), BLM-conditioned medium ($n = 5$) or OMC-conditioned medium ($n = 4$). Each symbol represents an individual donor (Mean±SEM; one-way ANOVA and Tukey's multiple comparisons tests). **b** Heatmap of the indicated soluble factors in conditioned media (CM) from 2D-cultured Fibroblasts and BLM cells, as well as from 3D-cultured OSC and OMC, as assessed by Luminex-based bead immunoassay and dedicated PGE2 ELISA. **c** Frequency of CD14$^+$ DCs in cDC2s cultured in DC medium ($n = 4$) only, in the absence or presence of recombinant human (rh) factors ($n = 4$). Each symbol represents an individual donor (mean ± SEM; two-way ANOVA and Tukey's multiple comparisons tests). **d** Fold change of the percentage of CD14$^+$DCs relative to their frequency in DC medium. IL-6 blockade attenuates the cDC2s conversion induced by rhIL-6 ($n = 4$), BLM-c.m. ($n = 5$) and OMC-c.m ($n = 2$). Each symbol represents an individual donor (mean ± SEM; two-tailed paired $t$ tests) ($p = 0.0024$). **e** Flow cytometry analysis shows phenotype of cDC2s and CD14$^+$ DCs isolated from OMCs and conventional tumour-cDC2 co-cultures (2D). Overlaid histograms for the indicated markers, from one representative experiment out of four, are shown. Colour legends indicate: cDC2s (orange), CD14$^+$DCs (green). Statistical significance was annotated as follows: *$p <$ 0.05, **$p <$ 0.01, ***$p <$ 0.001. Source data are provided as a Source data file.

study of tumour and immune cell migration and interaction. Remarkably, the changes we observed on myeloid cell subsets, including those isolated from melanoma patients, cultured in OMCs (either allogeneic or autologous) did not require the addition of exogenous growth factors or cytokines. Thus, confirming the suitability of the human decellularized dermal scaffold to maintain tumour and immune representativeness of the original lesion. Further functional experiments would demonstrate the employability of the scaffold for advanced T-cell experimenting. Potentially, the decellularized human tissue may represent a better alternative to animal-derived scaffolds that may act as a source of foreign antigen[40].

The intra-tumoural infiltration and activation of myeloid cells, such as DCs, has clinical relevance across different tumour types[18]. Thus, understanding the mechanisms that regulate the fate of individual DC subsets within the TME will be pivotal to define strategies that revert DC immunosuppression while simultaneously enhancing their activation[41]. In this regard, the behaviour of antigen-presenting cells in organotypic tumour cultures has never been addressed. We therefore validated the OMC by following the response of circulating human cDC2s within the reconstructed TME. cDC2s are phenotypically defined as CD1c$^+$CD14$^-$ and characterized by the ability to stimulate cytotoxic T-cell responses, exemplified by their use in DC vaccination protocols[27]. We documented the interaction of cDC2s with stromal and melanoma cells within the OMC, and attested the local conversion of cDC2s into a distinct myeloid cell population. Of note, this tumour-induced myeloid subset, characterized by the decreased expression of the DC marker CD1c and the concomitant acquisition of monocyte/TAM markers (CD14, CD163, CD206 and MerTK), is consistent with CD14$^+$ DCs already described in literature in the context of cancer-related inflammation[24,25]. These findings argue in favour of the ability of the OMC to recapitulate in vivo-like phenomena. While the current literature indicates that CD14$^+$ DCs may arise from monocytes[19], we here provide evidence for a new route of melanoma-mediated conversion of cDC2s into CD14$^+$ DCs. Such cDC2s-inherent phenotypic plasticity is not counterintuitive, if we reason that single-cell-RNA sequencing recently revealed the unappreciated heterogeneity of blood cDC2s, isolated from healthy individuals. Indeed, a cell cluster characterized by a unique inflammatory gene signature, close to that of CD14$^+$ monocytes, has been identified within the CD14-negative cDC2 subset[42]. Functionally, the melanoma-induced CD14$^+$ DCs described here, express genes (such as SSP1, PTGS2, IL-6) previously associated with immunosuppressive myeloid cells[30,33] and, like monocytes and macrophages, have poor T-cell stimulatory ability. The evolution of conventional DCs into regulatory macrophage-like cells has been only proposed in murine models[43,44]; in this study we report the ability of human, naturally-occurring, mature cDC2s to be reprogrammed into CD14$^+$ DCs. Indeed, human in vitro studies investigating this

phenomenon were performed using monocyte-derived DCs (moDCs), which represent a poor surrogate of in vivo DCs[45,46].

IL-6 and PGE2 were identified as factors involved in the cDC2-to-CD14$^+$ DC conversion. This is in line with their ability to orchestrate immune myeloid modulation towards immunosuppression[30–32]. Interestingly, the PGE2 selective expression in OMC-CM likely results from the cross-talk between melanoma cells and fibroblasts within the reconstructed TME. This finding supports the hypothesis that the multicellular dimensionality of the TME plays a critical role in shaping cell phenotypes[5,7]. We here showed that the phenotype of tumour-induced CD14$^+$ DCs was not obtained in parallel intra-donor 2D co-culture experiments. In particular, although some expression of CD14 was observed, these cells failed to express the TAM markers, CD163 and MerTK. The in vitro differentiation of myeloid cells to acquire TAM traits usually requires ≥3 days[47,48]. However, we believe that this rapid cDC2s conversion (within 3 days) could be explained by the fact that tumour and immune cells are not added in culture simultaneously, as usually performed in classical 2D co-culture experiments, but instead immune cells are exposed to an already tumour-conditioned skin microenvironment. This certainly leads to a stronger tumour-mediated effect and more closely resembles the natural cDC2s entry into tumour tissues in vivo. Intriguingly, CD14$^+$ DCs developed within the organotypic melanoma cultures shared important phenotypic similarities with CD14$^+$ DCs infiltrating human melanoma lesions. Moreover, they express higher levels of TAM-related markers (CD206, MerTK and CD163), compared to blood circulating CD14$^+$ DCs, we previously linked to lower immunological response to DC-based vaccines in melanoma patients[23,26]. These results demonstrate the added value of the OMC over classical 2D co-culture cell systems. Importantly, as highlighted by the time-course analysis, the cDC2 conversion occurring in the OMC at 48 h was even more profound after 72 h, indicating how critical the immunomodulatory effect of the TME is on the phenotype of myeloid cells. Our data indicate that the decellularized human-derived dermis repopulated with stromal cells, not only provides the correct extracellular space and guidance for cells to engage in dynamic cell–cell and cell–matrix interactions, but also offers the mechanical tissue stiffness as observed in vivo, and which likely contributes to TAM differentiation[49]. Ex vivo tumour-infiltrating CD14$^+$ DCs are characterized by the expression of the T-cell inhibitory molecule PD-L1. Interestingly, the upregulation of this key immune escape molecule was higher in tumour-educated myeloid cells within the OMC, and progressively increased over time, compared with those developed in classical monolayer co-cultures. To this end, it would be of interest to use the model to assess the modulation of immune checkpoint inhibition.

Together, our results show that the OMC culture system offers a number of advantages over two-dimensional culture systems. It facilitates studying development of tumours in more complex 3D

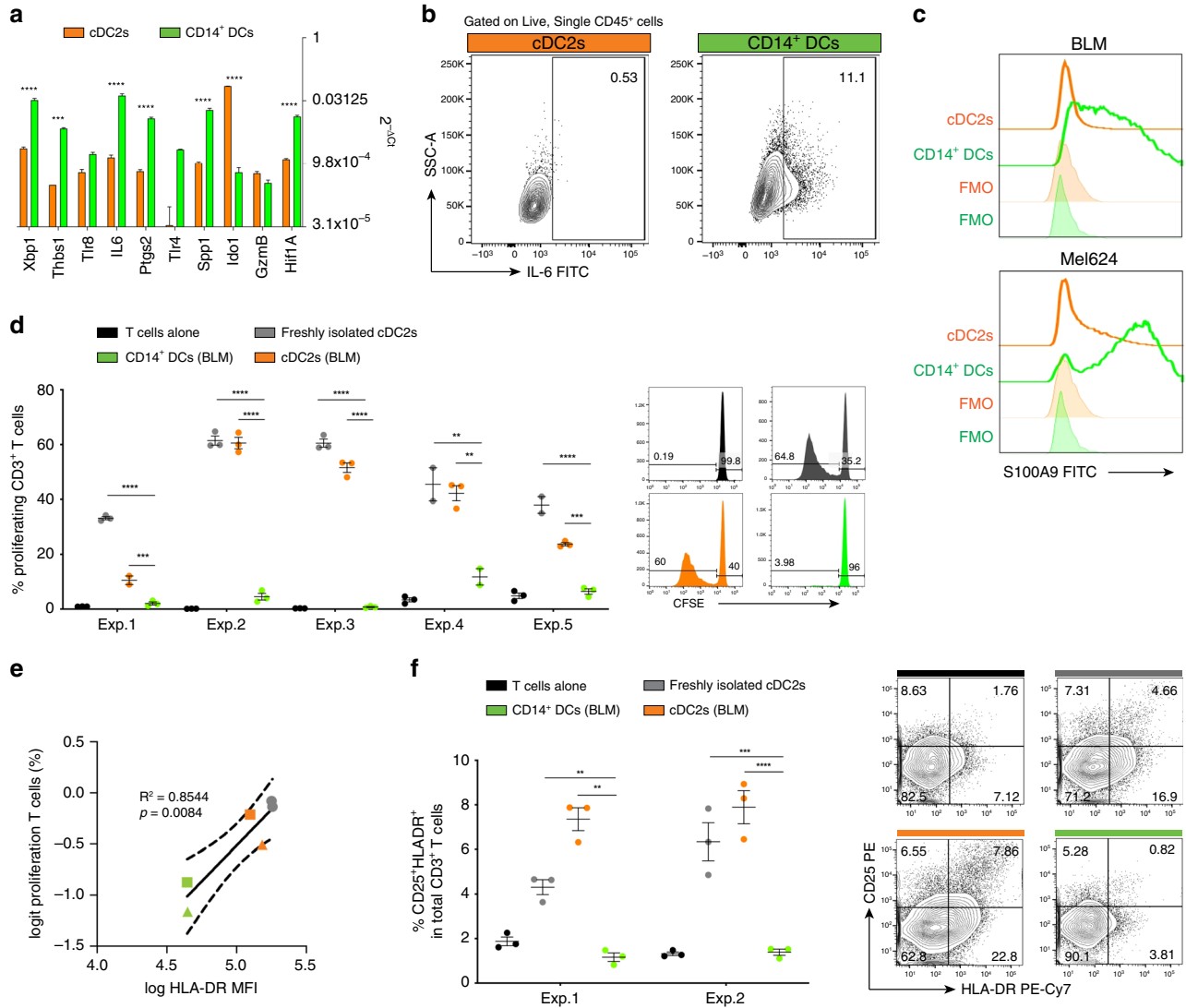

**Fig. 6 Functional characterization of cDC2s and CD14+ DCs isolated from human OMC. a** Two days after total cDC2s injection, BLM-OMCs were digested and CD1c+CD14− (cDC2s) and CD1c+CD14+ (CD14+DCs) subsets FAC-sorted for RNA extraction and molecular characterization by qRT-PCR. Gene expression levels ($2^{-\Delta Ct}$) for the indicated genes in cDC2s and CD14+DCs. Reported values for each gene are the means of $n = 3$ biological replicates (mean ± SEM; two-way ANOVA and Sidak's multiple comparisons test). ACTB was used as internal reference. Colour legends indicate: cDC2s (orange), CD14+DCs (green). **b** IL-6 production assayed by intracellular cytokine staining in BLM-educated cDC2s, upon stimulation with LPS (1 μg/mL) for 6 h. cDC2s and CD14+DCs were identified and gated based on CD1c and CD14 expression in CD45+ live single cells. Representative dot plots are shown. Numbers indicate the percentage of gated cells. **c** Intracellular protein expression of S100A9 in tumour-educated cDC2s and CD14+DCs. Representative GeoMFI histograms for BLM and Mel624 are shown. **d** Proliferation of allogeneic CD3+ T cells 5 days after co-culture with FAC-sorted cDC2s and CD14+DCs. Scattered dot plots (five independent experiments, in triplicate (Exps. 1-2-3) and duplicate (Exps. 4-5) samples; mean ± SEM; one-way analysis of variance (ANOVA) and Tukey's multiple comparisons tests). Representative CFSE histogram plots from Exp. 2 are shown; numbers indicate the percentage of gated cells. **e** Biplot of HLA-DR expression and induced allogeneic T-cell proliferation in Exps. 4 and 5. Correlation assessed by linear regression. **f** Autologous T cells were co-cultured with cDC2s and CD14+DCs for 5 days and then stained for HLA-DR, CD25, CD3, CD8 and live/dead marker. Percentage of activated CD25+HLA-DR+ cells in total CD3+ T cells is reported. Scattered dot plots (two independent experiments, in triplicate samples (mean ± SEM; one-way analysis of variance (ANOVA) and Tukey's multiple comparisons tests). Representative dot plots from Exp. 1 are shown. Statistical significance was annotated as follows: **$p < 0.01$, ***$p < 0.001$, ****$p < 0.0001$. Source data are provided as a Source data file.

tissues, such as skin, and at the same time investigating the effect of interactions with components of the immune system, such as DCs. These standardized OMC cultures can be used to get more detailed insights in how tumours can suppress the immune system.

We are well aware that, like most organotypic culture systems, our OMC is limited by the absence of blood vessels and lymphatics, and thus events such as angiogenesis and leukocyte extravasation cannot be mimicked. In order to overcome this caveat, attempts have been made to engineer collagen matrices containing a microvasculature network of endothelial cells, or to integrate microfluidics with tissue engineering to mimic the function of native microvessels[50,51]. Those approaches are still in their early infancy and further development is needed[52,53]. Within the OMC, DCs engage in dynamic interactions with tumour cells that are distributed in the reconstructed TME. It would be interesting to define the migratory nature of DCs towards tumour cells, whether this is a stochastic process or rather driven by a chemotactic gradient, secreted by the tumour. To support this kind of analyses, the OMC culture as described

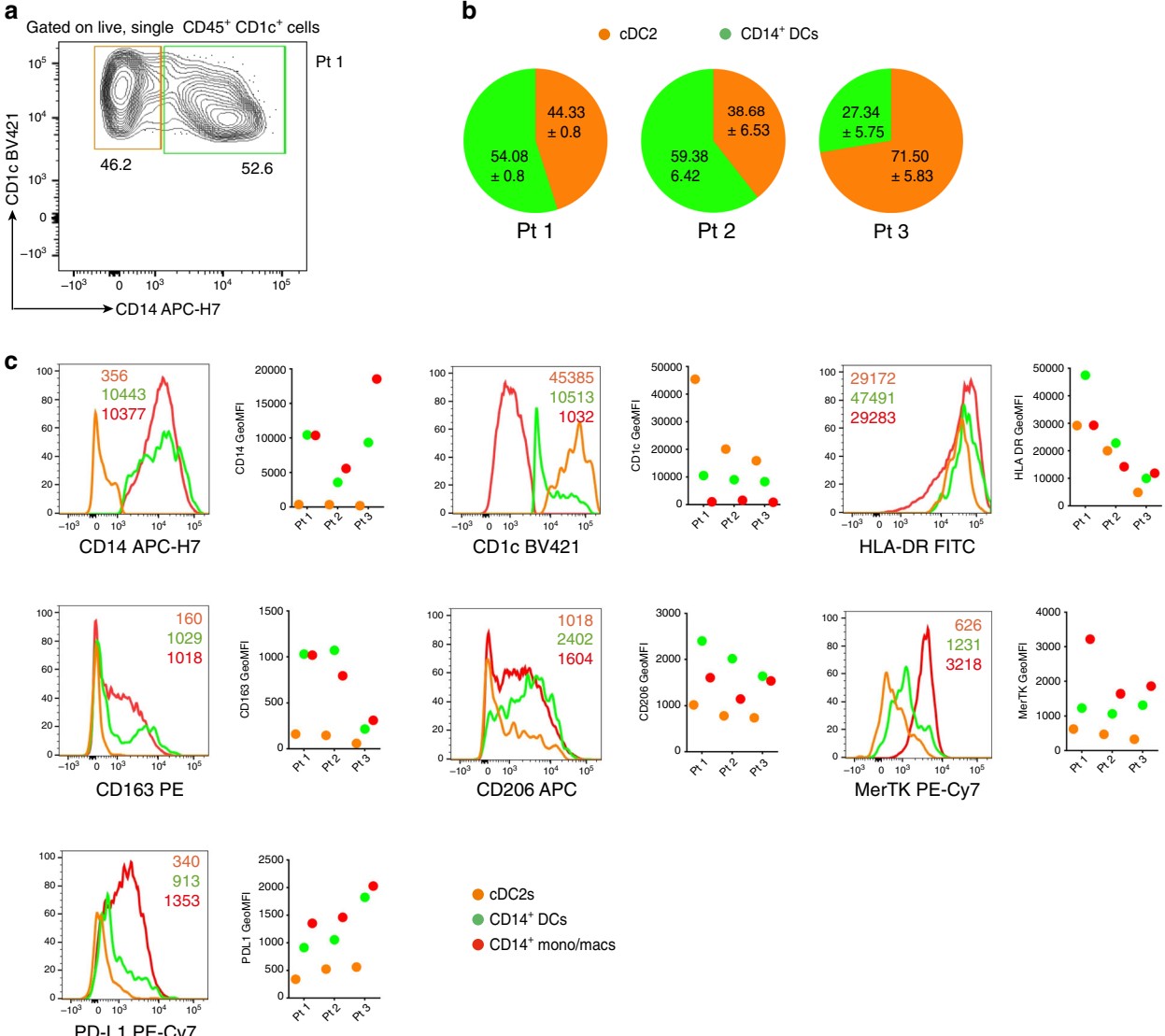

**Fig. 7 CD14⁺DCs in melanoma lesion phenotypically resemble those recapitulated in human OMC.** Human metastatic melanoma lesions were digested and stained (n = 3). **a** CD1c⁺CD14⁻ (cDC2s) and CD1c⁺CD14⁺ (CD14⁺ DCs) were identified and gated based on CD1c and CD14 expression in CD45⁺ CD1c⁺ live single cells. A representative contour plot from one patient (Pt. 1) is shown. **b** Pie charts report percentages (mean ± SD) of cDC2s and CD14⁺ DCs across different melanoma patients (n = 3, in triplicate samples; mean ± SD). **c** Graphs showing GeoMFI's of the phenotypic analysis of cDC2s, CD14⁺ DCs and CD14⁺ mono/macs defined within intra-tumoural CD45⁺CD11c⁺ live single cells from melanoma patient samples. Representative histogram plots for each indicated marker are shown. Colour legends indicate: cDC2s (orange), CD14⁺ DCs (green), CD14⁺ mono/macs (red). Source data are provided as a Source data file.

here would require further optimization. In particular, to ensure that location of both tumour cells (source of the chemotactic gradient) and DCs (responders) is better controlled. For example, by precise injection of the tumour and immune cells on opposite sides of the OMC.

In the present study we describe the generation, characterization and application of a human OMC, encompassing both stromal and immune compartments. Importantly, in contrast to patient-derived organotypic tumour cultures, our system does not aim to assess pre-existing immune infiltration[54], but rather has to be viewed as a fully tuneable melanoma environment for the in-depth investigation of early events that regulate tumour-immunological mechanisms. Overall, we believe that the proposed OMC will contribute to the identification of candidate genes and molecules that contribute to immune escape processes. Ultimately, this might even lead to the design of

novel therapeutic targeted approaches addressing the melanoma microenvironment.

## Methods

**Cell culture, transduction, stable cell line development**. Human melanoma cells (BLM, BLM-GFP, Mel624 and A375) were tested to be mycoplasma-free, authenticated by ATCC and maintained in Dulbecco's modified Eagle's medium (DMEM, Gibco), supplemented with 5% foetal calf serum and 5% $CO_2$ humidified air at 37 °C. The Lenti6/Block-iT-shScramble (GFP) vector was a kind gift of Prof. Peter Friedl (RIMLS, The Netherlands). The sequence of this construct does not match any known mammalian genes. BLM cells were infected with lentiviral vector and (10 μg/ml) polybrene and incubated at 37 °C, 5% $CO_2$, overnight. Then the medium was refreshed and cells were analysed after 72 h of treatment. A stable cell line was selected with 5 μg/ml blasticidin. Recombinant human factors IL-6 (Cell Genix), PGE2 (Pfizer) and CCL-2 (R&D Systems) were used in conversion experiments (all 20 ng/ml) in the presence or absence of anti-IL-6 (1 μg/ml, Invivogen).

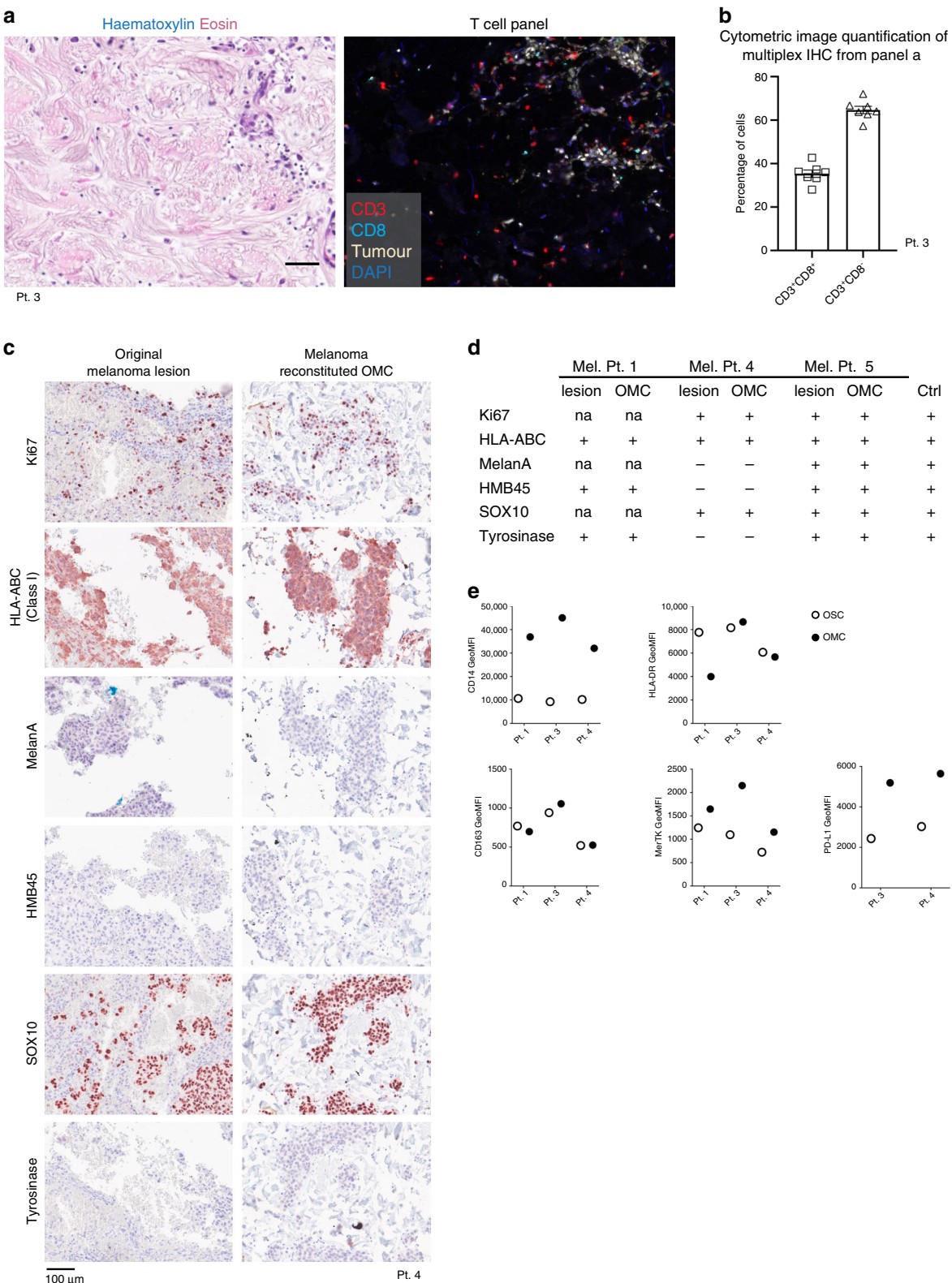

**Isolation of human blood immune cells**. Peripheral blood mononuclear cells (PBMCs) were isolated from buffy coats obtained from healthy volunteers (Sanquin) and purified via centrifugation over a Ficoll density gradient (Axis-Shield) in SepMate tubes (Stemcell technologies). cDC2s (CD1c⁺) cells were purified with magnetic cell sorting (MACS) from healthy donor PBMCs using the CD1c (BDCA1) DC isolation kit (Miltenyi Biotec). To obtain a highly purified CD1c⁺ CD14⁻ population, a pre-depletion step of monocytes using CD14-MACS microbeads (Miltenyi Biotec) was included in the original manufacturer's protocol. DC purity was assessed by staining with primary directly labelled antibodies: anti-

CD1c, anti-CD14 and anti-CD20 Abs (Supplementary Table 1). Purity levels higher than 96% were achieved, determined by flow cytometry (Supplementary Fig. 6a). For live imaging experiments, cDC2s were pre-labelled with the membrane dye PKH26 (Sigma-Aldrich) according to manufacturers' protocols and resuspended in X-VIVO-15 medium (Lonza), supplemented with 2% human serum (HS, Sanquin). Autologous or allogeneic CD3⁺ T cells used in T-cell co-culture experiments were isolated using the Pan T-cell isolation kit (Miltenyi Biotec) following the manufacturer's instructions. Peripheral blood monocytes were isolated using CD14-MACS Microbeads (Miltenyi Biotec) according manufacturer's instructions.

**Fig. 8 Analysis of patient-derived OMC. a, b** Comparison of melanoma patient's cell suspension cultured in a de-epidermized human dermis versus ex vivo analysis. Two days after injection, microtissues containing the patient material were fixed and processed for paraffin embedding. **a** Multiplex IHC staining on tissue sections was performed using the following antibodies: CD3 (red), CD8 (light blue) and tumour marker (white). The latter consists of a mix of antibodies recognizing the melanoma antigens: HMB45 (gp100), MelanA (MART-1), Tyrosinase and SOX10. Representative H&E stain and multicolour composite pictures for T-cell-related antibodies (T-cell panel) are shown ($n = 1$, Pt. 3). Scale bar, 50 μm. **b** Cytometric image manual quantification analysis of multiplex IHC sections ($n = 7$) shown in panel (**a**), to identify lymphocytes (tumour marker-CD3+CD8+ and Tumour marker-CD3+CD8− events). Mean ± SEM; two-tailed unpaired $t$ tests ($p < 0.0001$) (Gating strategy is shown in Supplementary Fig. 11). **c, d** Representative IHC images of tumour-specific characteristics in original melanoma lesion (left) and pt-derived OMC (right) ($n = 3$). Scale bars, 100 μm. **d** Overview of tumour-specific characteristics in original melanoma lesions and pt-derived OMCs. Ctrl is control melanoma tissue. **e** Graphs reporting the Geometric mean of fluorescence intensity (GeoMFI) of the indicated markers in CD14+ monocytes cultured within OSCs or pt-derived OMCs ($n = 3$; except for PD-L1: $n = 2$). Source data are provided as a Source Data file.

**Isolation of cellular and extracellular matrix human skin components**. Generation of de-epidermized, decellularized dermis and isolation of human primary keratinocytes from human abdominal and breast skin, derived from donors who underwent surgery for abdominal wall and breast correction[55]. Briefly, human skin was incubated for 5–10 min in phosphate-buffered saline (PBS) at 56 °C to allow separation of the epidermis from the dermis. A de-epidermized human dermis was obtained by incubating the dermis for one month in PBS containing gentamicin (0.5 mg/ml; Life Technologies, Inc.) and antibiotic/antimycotic (Life Technologies, Inc.) at 37 °C. Continuous cycles of freezing and thawing ensured the depletion of all living cells in the dermis. Punches were prepared from this de-epidermized dermis using an 8-mm biopter, exposed to additional freezing/thawing cycles and frozen for further use. Keratinocytes were isolated from the epidermal layer by trypsin treatment for 16–20 h at 4 °C, and then cultured in the presence of irradiated (3295 cGy for 4.10 min) mouse fibroblasts 3T3 cells. 3T3 cells were seeded at a concentration of $3 \times 10^4$ cells per cm² in Greens medium, which consisted of two parts Dulbecco's modified Eagle's medium (Life Technologies), one part of Ham's F12 medium (Life Technologies), 10% foetal bovine serum (Hyclone), L-glutamine (4 mmol/L; Life Technologies, Inc.), penicillin/streptomycin (50 IU/ml; Life Technologies, Inc.), adenine (24.3 μg/ml; Calbiochem, San Diego, CA), insulin (5 μg/ml; Sigma, St. Louis, MO), hydrocortisone (0.4 μg/ml; Merck, Darmstadt, Germany), triiodothyronine (1.36 ng/ml, Sigma), cholera toxin ($10^{-10}$ mol/L, Sigma). The next day keratinocytes were added at a concentration of $5 \times 10^4$ cells per cm². Keratinocytes-3T3 cells co-cultures were maintained for 3 days in Greens medium. Thereafter, medium was replaced by Greens medium containing epidermal growth factor (EGF, 10 ng/ml; Sigma), cells were expanded until 90% confluence and stored in the liquid nitrogen. Keratinocytes were used at passage one or two. Adult human dermal fibroblasts were purchased by ATCC and maintained in Fibroblast medium (3:1 DMEM: Ham's F12 Nutrient Mixture, Gibco) supplemented with 10% FCS, and expanded until 80% confluence. Passages three to nine were used for the experiments.

**Generation of the human organotypic skin melanoma culture**. To generate the OMC, 8 mm punch biopsy of de-epidermized, decellularized dermis was placed carefully basal membrane side down on a transwell insert in a 24-well plate (24-wells ThinCert, Greiner Bio-One) and $0.25 \times 10^6$ fibroblasts were seeded onto the reticular dermal side (opposite to papillary side and basal membrane) of a de-epidermized dermis, via centrifugal force (day 1), in Fibroblast medium (3:1 DMEM: Ham's F12 Nutrient Mixture, Gibco) supplemented with 10% FCS. The plate was centrifuged for 1 h at ~300 × $g$. After centrifugation, the de-epidermized dermis was maintained in Fibroblast medium for 2 days at 37 °C, to allow cellular proliferation and distribution through the structural collagen bundles and elastin fibres. After 2 days of culture, the repopulated de-epidermized dermis was turned basal membrane side up in the transwell insert and keratinocytes were seeded together with melanoma cells (BLM, Mel624 or A375) onto the papillary side of the dermal scaffold (day 3). The OMC was cultured submerged for 3 days in PONEC medium containing 5% serum (PONEC 5% medium), to allow proliferation of keratinocytes and tumour cells (day 6). PONEC 5% medium consists of two parts Dulbecco's modified Eagle's medium (Life Technologies, Inc.), one part of Ham's F12 medium (Life Technologies, Inc.), 5% calf serum (Hyclone), L-glutamine (4 mmol/L; Life Technologies, Inc.), penicillin/streptomycin (50 IU/ml; Life Technologies, Inc.), adenine (24.3 μg/ml; Calbiochem, San Diego, CA), insulin (0.2 μmol/l; Sigma, St. Louis, MO), hydrocortisone (1 μmol/l; Merck, Darmstadt, Germany), triiodothyronine (1.36 ng/ml, Sigma), cholera toxin ($10^{-10}$ mol/l, Sigma), ascorbic acid (50 μg/ml; Sigma). Thereafter, the OMC was shifted to the air–liquid interface and cultured for 11 days in PONEC medium without serum, supplemented with keratinocyte and epidermal growth factors (PONEC 0% medium), during which a fully differentiated epidermal layer is formed (day 17). PONEC 0% medium consists of two parts Dulbecco's modified Eagle's medium (Life Technologies, Inc.), one part of Ham's F12 medium (Life Technologies, Inc.), L-glutamine (4 mmol/L; Life Technologies, Inc.), penicillin/streptomycin (50 IU/ml; Life Technologies, Inc.), adenine (24.3 μg/ml; Calbiochem, San Diego, CA), L-serine (1 mg/ml; Sigma), L-carnitine (2 μg/ml, sigma), bovine serum albumin lipid mix (palmitic acid 25 μmol/l; arachidonic acid 7 μmol/l; linoleic acid 15 μmol/l; vitamin E 0.4 μg/ml; all from Sigma), insulin (0.1 μmol/l; Sigma, St. Louis, MO),

hydrocortisone (1 μmol/l; Merck, Darmstadt, Germany), triiodothyronine (1.36 ng/ml, Sigma), cholera toxin ($10^{-10}$ mol/L, Sigma), ascorbic acid (50 μg/ml; Sigma), keratinocyte growth factor (5 ng/ml, Sigma), epidermal growth factor (2 ng/ml, Sigma). At this point, the OMC was conditioned for 2 days with X-VIVO-15 medium (Lonza) supplemented with 10% human serum (Sanquin), to sustain survival of immune cells in the reconstructed microenvironment (day 19). Ex vivo culture of immune cells, which can be freshly isolated from peripheral blood circulation or separated from the tissue they infiltrate, can be a critical procedure due to the fragile nature of those cell types when dissected from their original microenvironment. At day 19, $0.1$–$0.5 \times 10^6$ cDC2s were microinjected into the dermis using a microneedle array system (NanoPass Technologies Ltd, Israel). The microneedle array holder was connected to a manual syringe pump (1 ml, BD). Microinjection was performed using air pressure (Supplementary Movie 1). Care was taken not to allow the microneedles to rupture the regenerated skin. Immunocompetent OMCs were cultured for an additional 2 days in X-VIVO-15 medium plus 10% human serum. For experiments that assessed DC phenotype and function, melanoma cells were incorporated (1) together with Fbs onto the dermal scaffold (day 1), and (2) co-seeded with keratinocytes onto the basal membrane (day 3). The double seeding procedure of melanoma cells was performed to guarantee efficient melanoma conditioning of the reconstructed microenvironment, in a way that would more closely recapitulate in vivo tumour-associated tissues.

**Patient material**. Tumour specimens were collected from metastatic melanoma patients who underwent a surgical resection with palliative intent. The following cases were analysed: Patient 1, dermal and subcutaneous metastasis axilla; Patient 2, adrenal metastasis; Patient 3, liver metastasis; Patient 4, subcutaneous metastasis and Patient 5, brain metastasis. The study protocol (CMO dossier-number 2016-2758) was approved by our Institutional Review Board, Centrale Commissie Mensgebonden Onderzoek (CCMO), and written informed consent was obtained from all patients.

**Tumour dissociation**. Melanoma cell suspensions were obtained from tumour samples by enzymatic and mechanic digestion using the gentleMACS Dissociator (Miltenyi, Bergisch-Gladbach, Germany). Briefly, tumour specimens were minced under sterile conditions into small pieces and digested over 1 h at 37 °C following the gentleMACS Dissociator protocol (Miltenyi). The resulting cell suspension was filtered through a 70-μm mesh (BD Biosciences, San Jose, CA, USA), the red blood cells were lysed, and the cell suspension was washed with RPMI. Cells were stored in liquid nitrogen until use. The same procedure was used for processing OMCs and OSCs, for the isolation and analysis of immune cells.

**Chromogenic IHC and manual digital cell density analysis**. Slides of 4- or 6-μm thickness were cut from formalin-fixed, paraffin-embedded (FFPE) primary melanoma tissue blocks, OMCs and dermal scaffold before and after decellularization. Haematoxylin-Eosin (HE) and Elastica van Gieson histological stainings were performed according to standard protocols. For chromogenic immunohistochemistry, antigen retrieval was performed by rehydrating and boiling the slides in either Tris-EDTA buffer (pH 9; 643901; Klinipath) for 10 min or in pronase (0.1% protease XIV, P5147-5G; Sigma) for 6 min at 37 °C. Protein blocking was achieved with Normal Antibody Diluent (VWRKBD09-999; ImmunoLogic), followed by wash in PBS on a rocking table at low speed. Primary antibodies (listed in Supplementary Table 2) were incubated for 1 h at room temperature. Subsequently, primary antibody detection and chromogenic visualization was performed with BrightVision poly-HRP (DPVR110HRP; ImmunoLogic) for 30 min at room temperature and DAB (VWRKBS04-999; ImmunoLogic) or NovaRed (SK-4800; Vector) for 7 min at room temperature. After dehydration, slides were counter-stained with haematoxylin for 1 min and enclosed with Quick-D mounting medium (7281; Klinipath). A selection of 5 representative original brightfield images per HE-stained section was loaded into the open source-imaging platform, Fiji (ImageJ 64 Bit for Windows), and used for blinded histopathologic evaluation of cell nuclei density, using the specific cell counter plug-in. Relative vascularity was determined in three representative CD31-stained tissue sections, by CD31-based

tissue segmentation using advanced Image Analysis software (inform 2.4.1; PerkinElmer).

**Fluorescent multiplex immunohistochemistry.** Formalin-fixed, paraffin-embedded (FFPE) OMCs were sectioned following a perpendicular orientation with respect to the direction of the epidermal layer (Supplementary Fig. 4a). Tissue sections (6-μm thickness) cut at a distance of ~200 μm were stained for fluorescent multiplex IHC. Immunofluorescent visualization was performed with the Opal seven-colour IHC Kit (NEL801001KT; PerkinElmer) containing the fluorophores: DAPI, Opal 520, Opal 540, Opal 570, Opal 620, Opal 650 and Opal 690[56]. Slides were boiled in Tris-EDTA buffer for antigen retrieval and removal of Ab-TSA complexes. Primary antibodies used are listed in Supplementary Table 3. A cocktail of monoclonal antibodies (moAbs, also called Tumour marker) directed towards two melanoma-associated antigens was added to each panel, including anti-tyrosinase and anti-SOX10. After Ab staining, slides were counterstained with DAPI for 5 min and enclosed in Fluoromount-G (0100-01; Southern Biotech).

**Tissue imaging and quantitative digital analysis.** Whole tissue slides were imaged using Vectra Intelligent Slide Analysis System (Version 3.0.4, PerkinElmer Inc.). This imaging technology combines imaging and spectroscopy to collect entire spectra at every location of the image plane. Regions of interest (ROI) were selected using Phenochart (Version 1.0.9, PerkinElmer Inc.). Images of single stained tissues for each reagent were used to build spectral libraries of the single dyes by using the inForm Advanced Image Analysis Software (Version 2.4.1, PerkinElmer Inc.). These spectral libraries were used to unmix the original multispectral images obtained with the Vectra imaging system (Supplementary Fig. 4c), to obtain an accurate and specific quantification of the Cleaved-Caspase-3 negative, CD45 positive (Cl. Cas3−CD45+) signal. A selection of 10–20 representative original multispectral images was used to train the inForm Advanced Image Analysis Software (Version 2.4.1, PerkinElmer Inc.) for quantitative image analysis (tissue segmentation, cell segmentation and phenotyping)[56,57]. All settings applied to the training images were saved within an algorithm allowing batch analysis of multiple original multispectral images of the same tissue. Phenochart (Version 1.0.9, PerkinElmer Inc.) was used to select the areas for analysis; this consisted of the entire reconstructed tissue. Both background area and tissue area (region of interest or ROI; Supplementary Fig. 4b) were defined by trainable tissue segmentation, based on morphological features and expression of DAPI (for efficient cell segmentation) and tumour markers (for discrimination of areas containing tumour cells). A region of disinterest (ROD) was manually drawn in a qualitative manner over the stratum corneum, dermal border and injection site in order to reduce the auto-fluorescent signal caused by the structural characteristics of those regions. To appreciate the cell distribution patterns at low resolution, we clustered the cell positions using hierarchical mean linkage clustering, with a distance threshold of 150 mm (i.e., clusters whose centres were more than 150 mm apart were not joined). Single-cell-based information was saved to a file format compatible with flow and image cytometry data analysis software FlowJo (Version 10, Treestar).

**Time-lapse image acquisition and analysis.** All time-lapse experiments were performed using multiparameter multiphoton microscopes (TriMScope-II, LaVision BioTec, Bielefeld, Germany), equipped with water objectives (Olympus XLPLN25XWMP2 1.05 NA and Nikon MRD77220 25×1.1 NA), on a temperature-controlled stage (recorded temperature in samples 35.4 °C). 4D time-lapse recordings were acquired by sequential scanning with 950 nm (eGFP and PKH26) and 1090 nm (SHG) excitation. Emission was bandpass filtered (525/50 eGFP and SHG, 620/60 PKH26) and detected with GaAsP photomultiplier detectors (Hamamatsu, H7422A-40). Areas of interest were imaged with 120 s time interval between individual scans over a 2-h period, starting ~4 h after injection of pre-labelled cDC2s. Consecutive images were acquired with a step-size of 5 μm. Images were analysed using the open source-imaging platform, Fiji (imageJ 64 Bit for Windows). Drifts in time-lapse recordings were corrected using the Descriptor-based series registration (2d/3d + t) plug-in ref. Gaussian filtering (0.5) was applied to all images prior to analysis. If necessary, images were scaled and adjusted for brightness and contrast to enhance visualization.

Step-by-step image processing and analysis of DC cell morphology is depicted in Supplementary Fig. 5a–c. Five fields-of-view (FOVs) from two independent experiments were analysed, over 50-min time-lapse two-photon microscopy movies. In each FOV, the tumour area represented 10–20% of the total imaging area. Maximum intensity projections of three consecutive z-stacks, for a total z-size of 10 μm were generated. Regions of interest (ROIs) were drawn to define DCs in proximity to tumour (distance <100 μm, Proximal) and DCs distant from tumour (distance >100 μm, Distal). A semi-automated analysis of cell shape parameters within either proximal or distant ROIs was performed. Cell clusters were excluded from the analysis based on size. Roundness was used as a measurement of how cell shape is close to a circle (whereby, 1 = circular cell, 0 = elongated cell), and to indicate degree of cell protrusion. In our analysis, a roundness factor of ~0.6 was calculated in cells with a typical hand-mirror-shape phenotype (Supplementary Fig. 5b). DC morphology was assessed in time-lapse images with a time-lapse interval of 10 min. Results from two experiments are shown in Supplementary

Fig. 5c. The image processing and quantification steps of DCs (magenta) interacting with tumour cells (BLM-GFP, green) in OMCs, are depicted in Supplementary Fig. 5d. Five fields-of-view (FOVs) from two independent experiments were analysed, with a time-lapse interval of 2 min. In each FOV, the area occupied by tumour cells represents 10–20% of the total imaging area (Supplementary Fig. 5e). Maximum intensity projections of three consecutive z-stacks, for a total z-size of 10 μm were generated. The semi-automated quantification of DC-tumour interactions is based on the measurement of mean gray values of GFP pixel intensities (tumour) that overlay with DC selections. Resulting mean gray values were corrected by background subtraction. The percentage of DCs interacting with tumour cells over total DCs was reported for each area of interest.

**Flow cytometry and fluorescent-activated cell sorting (FACS).** A complete list of antibodies used in the study is reported in Supplementary Table 1. Dead cells were identified using Fixable Viability Dye eFluor® 506 or 450 (Affymetrix, eBiosciences) and excluded from the analysis. For surface staining: cells were incubated in 2% human serum for the blocking of non-specific antibody binding to receptors (10 min, 4 °C) and subsequently stained with directly labelled primary antibodies (30 min, 4 °C) (Supplementary Table 1). For IL-6 intracellular staining: cells were stimulated with LPS (1 μg/mL, Sigma) for 6 h and GolgiPlug (BD Biosciences) was added in the last 5 h of the assay. After staining of surface markers (using directly labelled anti-CD45, anti-CD1c and anti-CD14 Abs), cells were washed, fixed and permeabilized with Cytofix/Cytoperm buffer (BD Biosciences) and stained with FITC-labelled anti-IL-6 (BioLegend). Acquisition was performed on a FACSVerse flow cytometer (BD Biosciences) and FlowJo analysis software (Treestar) was used for data analysis. Fluorescent-activated cell sorting (FACS) was performed using the ARIA SORP (Becton Dickinson, Franklin Lakes, NJ). Anti-CD45, anti-CD1c and anti-CD14 sterile antibodies were used. Briefly, 2 days post-injection tumour-free OSCs and OMCs were digested, under sterile conditions, and the cell suspensions filtered, stained and cells were sorted with >98% purity.

**Mixed lymphocyte reaction and activated T-cell phenotype.** The ability of cDC2s (CD1c+CD14−) and CD14+ DCs (CD1c+CD14+), isolated from OSCs and OMCs, to induce T-cell proliferation was tested in a mixed lymphocyte reaction (MLR). A total of 3 × 10⁴ CFSE-labelled unstimulated allogeneic CD3+ T cells from a healthy donor were seeded in a round bottom 96-well/plate and 15,000 (1:2 ratio), 6000 (1:5 ratio) or 0/myeloid cells were added. The percentage of pro-liferated T cells was analysed at day 5 after staining with anti-CD3 and anti-CD8 antibodies. For assessing the myeloid subsets stimulatory potential on autologous T cells, 1 × 10⁵ autologous unfractionated CD3+ T cells were cultured with 10,000 (1:10 ratio) or 0/myeloid cells. After 5 days, T cells were stained with anti-CD3, CD8, CD25 and HLA-DR (see Supplementary Table 1) and acquired at the FACSVerse (BD Biosciences).

**Gene expression analysis.** Total RNA was extracted from FAC-sorted immune cells using Trizol (Invitrogen), and cDNA was generated using the SuperScript II Reverse Transcriptase Kit (ThermoFisher Scientific). cDNA was then used as a template for messenger RNA (mRNA) amplification using the TranscriptAid T7 High Yield Transcription Kit (Thermo Scientific). Amplified (aRNA) was purified using Agencourt RNAClean XP beads (Beckman Coulter, Brea, CA) prior to Nanodrop quantification. 500 ng/mL of aRNA were used for reverse transcription with Superscript II (ThermoFisher Scientific). The resulting cDNA was used as a template. qPCR using FastStart SYBR Green Master (Roche Diagnostic GmbH) was performed on a CFX96 real-time cycler (Bio-Rad). The 2−ΔCt method, in which Ct represents the threshold cycle, was applied. Samples were run in triplicate. Relative gene expression was determined by normalizing the gene expression of each target gene to β-actin (ACTB). Primer sequences are listed in Supplementary Table 4.

**Quantification of soluble factors detection in supernatant.** Forty-five soluble factors in 48 h cultured supernatant from fibroblasts, BLM, OSC and OMC were quantified by using either the Luminex-based bead multiplex immunoassays (Millipore) or a dedicated ELISA for PGE2 (R&D Systems) according to the respective manufacturers' instructions. Heatmap showing concentrations of soluble factors among conditions were generated with Rstudio (Version 1.2.1335, RStudio Team 2018), using log2-transformed cytokine concentrations[58].

**T-cell analysis of PBMCs injected in the OMC.** Peripheral blood mononuclear cells (PBMCs) were isolated from buffy coats obtained from healthy volunteers (Sanquin) and purified via centrifugation over a Ficoll. OMCs containing BLM cells were generated as described above. Then, PBMCs were microinjected into the dermis of the generated OMCs (0.5 × 10⁶ cells/OMC) using a microneedle array system (kindly provided by Nanopass Technology Ltd, Israel). Immunocompetent OMCs were cultured for additional two days in X-VIVO-15 medium plus 10% human serum. In parallel, an aliquot of PBMCs was also cultured in a standard plate (2D) and maintained in the same culture medium. After 48 h the percentage of CD3+ and CD8+ cells was evaluated by flow cytometry. Briefly, dead cells were

identified using the Fixable Viability Dye eFluor® 506 (Affymetrix, eBiosciences) and excluded from the analysis. Then, cells were incubated in 2% human serum for the blocking of non-specific antibody binding to receptors (10 min, 4 °C) and subsequently stained with directly labelled primary antibodies to CD45, CD3 and CD8 (30 min, 4 °C) (Supplementary Table 1).

**Generation and analysis of patient-derived OMCs.** Patient-derived metastatic melanoma samples were processed as described above. The resulting single-cell suspensions were microinjected into a de-epidermized, decellularized human dermis using the microneedle array system (Nanopass Technology Ltd, Israel). The resulting patient-derived OMCs were cultured for two days in X-VIVO-15 medium plus 10% human serum. Then, microtissues containing the patient material were fixed and processed for paraffin embedding. Fluorescent Multiplex IHC staining on tissue sections was performed using the following antibodies: CD3, CD8 and tumour marker. The latter consists of a mix of antibodies recognizing the melanoma antigens, HMB45 (gp100), MelanA (MART-1), Tyrosinase and SOX10 (Supplementary Table 2). Prior to injection, flow cytometry was conducted on the same patient-derived samples using antibodies recognizing CD3, CD8 and CD45 (Supplementary Table 1). Dead cells were identified using the Fixable Viability Dye eFluor® 506 (Affymetrix, eBiosciences) and excluded from the analysis. For autologous experiments within the OMC the patient tumour suspension-derived CD45-negative fraction, containing tumour and stromal cells, was allowed to grow over two days within the dermal scaffold and CD14+ monocytes were isolated from peripheral blood and injected into the model similar to what described for cDC2s. Similarly, after 2 days the OMC-educated phenotype of monocytes was interrogated by flow cytometry on the digested OMC suspension.

**Statistical analysis.** Statistical analysis was performed using GraphPad Prism V6 and V8 (GraphPad Software Inc, San Diego, CA). Unless otherwise indicated, results are presented as mean±SEM in scattered dot plots. The significance between two groups was analysed by two-tailed paired or unpaired Student's $t$ tests or Wilcoxon matched-pairs signed-rank test. Multiple comparisons were performed using one-way or two-way analysis of variance (ANOVA) followed by multiple comparisons tests, as indicated in each figure legend. Statistical significance was annotated as follows: $*p < 0.05$, $**p < 0.01$, $***p < 0.001$, $****p < 0.0001$.

**Reporting summary.** Further information on research design is available in the Nature Research Reporting Summary linked to this article.

## Data availability

All the data supporting the findings of this study are available within the article and its supplementary information files and from the corresponding author upon reasonable request. The source data underlying Figs. 2d, 4b–e, 5a–d, 6a, d–f, 7b, c, 8b, e and Supplementary Figs. 1d, f, 5c, e, f, 7a, c, 8a, b, 9a–d are provided as a Source data file. A reporting summary for this article is available as a Supplementary Information file.

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

## Acknowledgements

The authors gratefully acknowledge Rob Woestenenk for valuable assistance and technical support, and Bas Pilzeker for generating the heatmap. For their assistance with cytokine quantification, we thank the MultiPlex Core Facility of the UMCU, the Netherlands. This work was supported by Dutch Cancer Society grants KWO2009-4402 and 10673. C.G.F. received the NWO Spinoza award and ERC Advanced Grant ARTimmune (834618), I.J.M. d.V. received Vici grant 91814655 from the Dutch research council. M.T. was supported by a FIRC (Fondazione Italiana per la Ricerca sul Cancro) Fellowship for abroad.

## Author contributions

S.D.B. and M.T. conceived the study, performed the experiments, analysed and interpreted the data, wrote the paper. G.F.v.W. performed the experiments, helped analysing the data and writing parts of the paper. A.v.D., A.B. and I.S. contributed to some experiments. M.B. arranged the collection of patient samples. K.V., M.G., G.J.B., A.H., A.V. and J.C.T. provided technical support. G.B., S.V.H., J.S. and I.J.M.d.V. contributed with critical feedback. J.H.W.d.W. provided clinical samples of metastatic melanoma patients. E.H.v.d.B. provided technical contribution and critical feedback. C.G.F. conceived the study, interpreted the data and critically reviewed the paper.

## Competing interests

The authors declare no competing interests.
