## [Peer Review File · Nature Communications]

Reviewers' comments:

Reviewer #1 (Remarks to the Author):

Authors have constructed an organotypic skin melanoma culture (OMC) by using keratinocytes, immune cells and fibroblasts in presence of tumor cells and claim this model is superior to PDX and previously established 3D models. Using the model, the authors have used live cell imaging by 2-photon microscope to monitor interaction of a subset of DC (cDC2) with tumor cells in tumor microenvironment. In the OMC model, the authors report that cDC2 convert into an immunosuppressive (CD14+ DC) phenotype in presence of the tumor cells. Such cells are unable to stimulate allogeneic lymphocytes in a mixed lymphocyte reaction (MLR).

Comments:

The authors describe a new technique to establish an in vitro organotypic skin melanoma culture model. Authors claim that their technique is superior as compared to other available in vivo PDX or similar 3D models that has been well described in the past. This manuscript appears to be very descriptive of the techniques used and does not explore the causes of conversion of a DC subset to an immunosuppressive CD14+ DC subtype under the tumor influence.

Specific comments:

1. In Fig 1c, authors claim that there are interactions of DCs, cancer associated fibroblasts and tumor cells. This is not very convincing. Authors have used primary fibroblasts in the culture and there is no description of the fibroblasts assuming the phenotype of CAFs. No phenotypic characterization of the fibroblasts isolated from the tumor microenvironment is described.
2. In Fig 3, what drives the DC subtype from cDC2 to CD14+ DC phenotype? What are the tumor factors that drives this phenotype? Do these DCs need cell to cell interactions? Are soluble factors enough to drive these phenotypic changes?
3. Authors claim that 48h are needed to convert cDC2 to phenotype to immunosuppressive phenotype. One needs to show time kinetics to record events as early as 8h to 72h.
4. In Fig 5d, authors need to report the expression of HLA-DR on the immunosuppressive DCs. Intensity of HLA-DR expression determines and drives an MLR reaction. One needs to know the mechanism of poor MLR response.

Minor points:

1. In Fig 5, authors have use CD25 as an activation marker. One should also use HLA DR and CD69 markers.
2. In Fig 1, authors need to provide scale bars instead of magnifications.

a. In the same Fig, the authors need to provide arrows to indicate different cell types in the organotypic cultures.

3. In Fig 3, how many DCs interaction with tumor cells during a given time?

Reviewer #2 (Remarks to the Author):

In this manuscript DiBalsio and co-authors described a novel human organotypic skin melanoma culture (OMC). They co-cultured decellularized dermis with keratinocytes, fibroblasts and immune cells in the presence of melanoma cells. Thus, they tried to generate a reconstructed tumor microenvironment. They demonstrated that the OMC is suitable and outperforms conventional 2D co-cultures for the study of dendritic cells dysfunction.

The presented experimental model is novel and potentially useful. Experiments are clearly described and conclusion are convincing. The potential usage of this system for the analysis of immune mechanisms is not clear at this moment since presented data are rather limited. However, publication of these results would open the possibility for scientific community to test it and thus would be desirable.

Reviewer #3 (Remarks to the Author):

In the study "The tumour microenvironment shapes dendritic cell plasticity in a human organotypic melanoma culture", Figdor and colleagues describe an in vitro model to study how the tumour microenvironment influences the immune system. This is interesting work with potential clinical significance. However, there are several issues that limit my enthusiasm and need to be addressed.

1. Only melanoma cell lines are used in the OMCs, which still makes it a very artificial model. It would be much more valuable if primary patient-derived melanoma cells as well as immune cells from the same patient can be implemented in the model.

2. Figure 1 describes OMC establishment. Yet, the OMCs are not well characterized. Are the melanoma cells still proliferating in the OMCs and how does this compare to primary tumour tissue?

The images in Figure 1C are not clear and primary tumour lesion should be included as comparison. Lastly, scale bars are missing.

3. To authors claim that DCs cells are migrating in the OMC by showing their distribution throughout the tissue sections 2 days after injection. However, to make such a claim the authors at least need to show DC distribution right after injection as well. Are they randomly migrating or moving towards tumour cells?

4. In line with my previous comment, it is unclear from Figure 3 if the interaction of DCs with melanoma cells and particles derived thereof is just a stochastic process, or whether the DCs are actively moving towards them. What percentage of DCs show a migratory “hand mirror shape” phenotype and are these typically the ones that are in close proximity of tumour cells?

5. The authors present the OMC model as a platform for exploring cancer-driven immune cell modulation. In Supp Fig 8, they demonstrate that human lymphocytes can be maintained in the OMCs as well. The manuscript would greatly benefit from some additional experiments demonstrating the applicability of OMCs to study immune modulation, such as DC-dependent cytotoxic T-cell stimulation and the effect of checkpoint inhibition on tumour cell killing.

Point-by-point reply to Reviewers' comments.

Changes are highlighted in yellow within the revised manuscript. Deleted parts are shown as crossed out text.

Reviewer #1:

Authors have constructed an organotypic skin melanoma culture (OMC) by using
keratinocytes, immune cells and fibroblasts in the presence of tumor cells and claim this
model is superior to PDX and previously established 3D models. Using the model, the authors
have used live cell imaging by 2-photon microscope to monitor interaction of a subset of DC
(cDC2) with tumor cells in tumor microenvironment. In the OMC model, the authors report
that cDC2 convert into an immunosuppressive (CD14⁺ DC) phenotype in presence of the
tumor cells. Such cells are unable to stimulate allogeneic lymphocytes in a mixed lymphocyte
reaction (MLR).

Comments:

The authors describe a new technique to establish an in vitro organotypic skin melanoma
culture model. Authors claim that their technique is superior as compared to other available
in vivo PDX or similar 3D models that has been well described in the past. This manuscript
appears to be very descriptive of the techniques used and does not explore the causes of
conversion of a DC subset to an immunosuppressive CD14⁺ DC subtype under the tumor
influence.

*As highlighted by the Reviewer, the submitted manuscript presents a strategy to establish a*
*novel in vitro Organotypic Melanoma Culture (OMC), suitable to explore cancer-driven*
*immunomodulation. We have applied this model to interrogate the interaction between*
*melanoma cells and myeloid immune cells occurring within the reconstructed 3D skin*
*melanoma tissue. As underlined in the manuscript, our model represents an important*
*technical advance in the field of 3D models for skin malignancies to study immunosuppressive*
*mechanisms [1-5]. In particular, we showed that the tumour microenvironment (TME)*
*corrupts cDC2 DCs and alters their stimulatory nature.*

*Within the revised manuscript, we aimed at better clarifying the mechanism driving this*
*conversion of cDC2s into an immunosuppressive (CD14⁺ DC) phenotype. We have carried out*
*additional experiments and included a detailed explanation on this in the reply to Reviewer 1-*
*Question 2 (please refer to pages 5 and 6 of this rebuttal letter).*

Specific comments:

1a. In Fig 1c, authors claim that there are interactions of DCs, cancer associated fibroblasts
and tumor cells. This is not very are convincing.

*We thank the Reviewer for this observation. We apologize if this was unclear and we would*
*like to underline that, Fig. 1 only describes the first part (from day 1 to day 17) of the*
*development of the OMC, containing keratinocytes, tumour cells and fibroblasts (please see*

schematics in Fig. 1a). Next, Fig. 2 depicts the incorporation of DCs within the fully-developed
 human OMC (as shown by the schematics in Fig. 2a).

 We agree with the Reviewer that Fig. 1c of the submitted manuscript did not allow an easy
 discrimination of fibroblasts (FAP⁺, magenta) interacting with tumour (Tumour⁺, yellow) cells.
 Therefore, we tested a panel of antibodies generally used for staining of fibroblasts in tissues.
 Details on fibroblast antibody selection are given below in the reply to Reviewer 1-Question
 1b. Amongst different antibodies, we chose the anti-FSP1 antibody for its high stroma-to-
 tumour selectivity [6,7], and used this for the staining of OMC tissue sections.

 Fig. 1c was modified to include a triple FSP1/tumour marker/DAPI staining of human
 melanoma (left) and in vitro generated OMC (right) tissue sections. We have included
 representative images of areas with both stromal-tumour clusters and dispersed fibroblasts
 (FSP1⁺ cells, magenta) and melanoma cells (Tumour⁺, yellow). “Pseudo-DAB” of isolated
 fluorescent channels of the same areas are shown and fibroblasts are indicated with arrows.

 The original FAP image has been moved to the Supplementary section (Suppl. Fig. 3a),
 together with the representative immunohistochemistry stainings of FAP and FSP1, performed
 on human melanoma lesions (Suppl. Fig. 3b). Text (page 6), Figure legend (page 26) and
 Supplementary Figure legend (page 5 of Supplementary material) have been modified
 accordingly.

 1b. Authors have used primary fibroblasts in the culture and there is no description of the
 fibroblasts assuming the phenotype of CAFs. No phenotypic characterization of fibroblasts
 isolated from the tumor microenvironment is described.

 In order to phenotypically characterize fibroblasts within the OMC, we assessed the expression
 of different fibroblast-associated markers (Vimentin, aSMA, FAP and FSP1) by
 immunohistochemistry. Details of the antibodies used can be found in the Rebuttal Table 1.

 Firstly, we compared their staining patterns both in healthy human skin and in two different
 human melanoma tissue biopsies (Rebuttal Fig. 1a). Both Vimentin and aSMA expression were
 not restricted to fibroblasts, showing staining positivity in melanoma cells and other stromal
 cells (including pericytes), respectively [8]. Furthermore, FAP (ab#1, Sigma Aldrich) showed
 strong non-specific staining at all concentrations tested, excluding issues that might be related

Marker	Clone	Cat#	Supplier	Dilution	Antigen Retrieval	Opal
Vimentin (1)	V9	NCL-L-VIM-V9	Novocastra	1/50	EDTA 10'	650
Vimentin (2)	V9	AM074-5M	Biogenex	1/4000	EDTA 10'	650
aSMA	1A4	a5228	Sigma Aldrich	1/4000	EDTA 10'	650
FAP (1)	Polyclonal	SAB4500839	Sigma Aldrich	1/100	EDTA 10'	650
FAP (2)	SP325	ab227703	Abcam	1/100	EDTA 10'	650
FSP1	CL0240	AMAB90599	Sigma Aldrich	1/2000	EDTA 10'	650

**Rebuttal Table 1. IHC Fibroblast antibody details**

to saturating antibody dilutions. Conversely, both FAP (ab#2, Abcam) and FSP1 antibodies
 specifically stained fibroblasts in both human skin and melanoma tissues (please see Rebuttal
 Fig. 1a). Secondly, we tested the suitability of the selected FAP (ab#2) and FSP1 antibodies to
 discriminate fibroblasts within the reconstructed OMC, and did not observe non-specific
 background staining on OMC tissue sections with either of the two antibodies (please see
 Rebuttal Fig. 1b). Together, we believe these observations support our choice for using both
 FAP and FSP1 for fibroblast identification in our system.

 **Rebuttal Figure 1. Fibroblast characterization and IHC antibody selection**
 **a**, Representative IHC images for Vimentin (ab#1 and #2), αSMA, FAP (ab#1 and #2) and FSP-1 in
 Healthy Skin (left) and two melanoma lesions (middle and right). DAPI (blue) indicates nuclei. Scale
 bars 100μm. **b**, Representative IHC staining for FAP (ab#2) and FSP-1 on human OSC and OMC lesions.
 DAPI (blue) indicates nuclei. Scale bars 100μm. **c**, Software-based quantification of FAP and FSP1
 expression, measured on stained sections from healthy skin and tumour lesions (top), and sections
 from OSCs and OMCs (bottom). Staining intensities of the given markers are expressed as Integrated
 density (in arbitrary units) and were measured using Fiji/Image J.

*Because the increased expression of these two fibroblast markers would be in favour of a*
*Cancer-Associated Fibroblast phenotype [6,8], we aimed at assessing their level of expression*
*in OSCs versus OMCs. Automated software-based quantification of staining intensities*
*(expressed as integrated density) did not suggest an increase in their expression associated*
*with the melanoma microenvironment (Rebuttal Fig. 1c). Therefore, we removed the*
*nomenclature “CAF” from the manuscript (pages 6 and 7). We believe that, given the cross-*
*talk between tumour and stromal cells within the reconstructed microenvironment (discussed*
*in reply to Reviewer 1-Question 2), it would be relevant to characterize melanoma-educated*
*fibroblasts in the OMC, as part of a new study.*

2. In Fig 3, what drives the DC subtype from cDC2 to CD14⁺ DC phenotype? What are the
tumor factors that drives this phenotype? Do these DCs need cell to cell interactions? Are
soluble factors enough to drive these phenotypic changes?

*We agree with the Reviewer that it is important to resolve the mechanism driving cDC2-to-*
*CD14⁺ DC conversion. This prompted us to carry out a series of new experiments to unravel*
*the factors responsible for the CD14⁺ DC phenotype.*

*To determine the role of potential soluble factors mediating tumour-induced myeloid cDC2-*
*to-CD14⁺ DC conversion, we exposed blood-derived cDC2 DCs to conditioned media from either*
*OMC (OMC-CM, generated using BLM melanoma cells, n=4) or BLM melanoma cells (BLM-*
*CM, n=5), cultured for 48h. Both OMC-CM and BLM-CM conditioned media induced conversion*
*of myeloid cells, as expressed by a significant increase in the percentage of CD14⁺ DCs (Fig.*
*5a). These findings indicate that soluble factors can indeed induce conversion independent of*
*cell-cell contact.*

*To further determine the origin of the secreted proteins in conditioned media, we performed*
*Luminex-based bead multiplex immunoassays and an ELISA assay (designed for PGE2). We*
*evaluated conditioned media from BLM melanoma cells (BLM-CM), fibroblasts (Fb-CM),*
*Organotypic Melanoma Culture (OMC-CM) and Organotypic Skin Culture (OSC-CM). The*
*heatmap derived from these results shows the distribution of soluble factors of the four*
*different cultures (Fig. 5b). Amongst those mediators, we focused our attention on three*
*immunomodulatory proteins that have been previously shown to potentiate tumour-induced*
*myeloid immunosuppression: CCL-2, IL-6 and PGE2 [9-11]. Thus, we tested the ability of these*
*selected factors (in the form of recombinant human (rh) proteins) alone or in combination, to*
*induce cDC2-to-CD14⁺ DC conversion, in vitro.*

*While the chemokine CCL-2 in conditioned media was not influenced by the presence of*
*tumour, and was found also in OSC-CM; by contrast, the inflammatory cytokine IL-6 was*
*clearly induced in cultures that contained tumour cells (BLM-CM and OMC-CM), but was lower*
*in Fb-CM and OSC-CM.*

*This was in line with our observation that only rhIL-6, but not rhCCL-2, drove the conversion of*
*stimulatory cDC2s towards immunosuppressive CD14⁺ DCs (Fig. 5c). The role of rhIL-6 in*
*modulating the generation of CD14⁺ DCs was confirmed by the significant reduction induced*
*upon addition of IL-6 blocking antibodies (Fig. 5d). Additionally, IL-6 blockade also partially*

*reduced the immunosuppressive modulation of cDC2 induced by either BLM-CM or OMC-CM.*
*This indicates that also other soluble factors may play a role. Therefore, we also studied PGE2.*
*The inflammatory mediator PGE2, was detected neither in tumour-free media (OSC-CM and*
*Fb-CM) nor in BLM-CM (Fig. 5b). However, we found that PGE2 was highly expressed in OMC-*
*CM, suggesting that the cross-talk between melanoma cells and fibroblasts within the*
*reconstructed TME induced PGE2 secretion. This finding further underlines the added value of*
*the organotypic melanoma model, where multiple cell types are present, to study cellular*
*cross-talk within a reconstructed TME.*

*In conclusion, we showed that PGE2 either alone or in combination with IL-6, produced within*
*the OMC only when tumour cells were present, were responsible for the conversion of cDC2*
*and potent inducers of CD14⁺ DCs (Fig. 5c). This notion is supported by our finding that the*
*PGE2-related COX-2 (Ptgs2) pathway is upregulated in CD14⁺ DCs (Fig. 6a). These new data*
*have now been incorporated in the manuscript in a newly generated Fig. 5. Text (pages 9-10*
*and 14) and Figure legend (pages 27-28) have been changed accordingly.*

3. Authors claim that 48h are needed to convert cDC2 to phenotype to immunosuppressive
phenotype. One needs to show time kinetics to record events as early as 8h to 72h.

*To address the Reviewer's question, we monitored changes in the phenotype of cDC2s,*
*isolated from OMCs, at three different time points (8h, 48, and 72h), and compared the results*
*to those isolated from OSCs.*

*The frequency of CD14⁺ DCs increased (approximately 4-fold) already after 8h, further doubled*
*48h after injection, reflecting data of the original manuscript, and remained consistent*
*thereafter (Suppl. Fig 7a). The rapid accumulation of CD14⁺ DCs over time, was accompanied*
*with a prominent concomitant reduction of cDC2s in the TME (Suppl. Fig 7a).*

*Phenotypic analysis of CD14⁺ DCs showed a progressive upregulation of the*
*monocytic/macrophage markers, CD14 and CD163, as well as of the inhibitory molecule, PD-*
*L1. These changes were accompanied by a downregulation of CD1c and HLA-DR expression,*
*over time (Suppl. Fig 7b,c). Importantly, the tumour-induced modulation observed at 48h, is*
*more profound after 72h, indicating how critical the immunomodulatory effect of the TME is*
*on the phenotype of myeloid cells. The manuscript has been modified accordingly (page 9-10*
*and 15) and Supplementary Figure legend (page 7 of Supplementary material).*

4. In Fig 5d, authors need to report the expression of HLA-DR on the immunosuppressive DCs.
Intensity of HLA-DR expression determines and drives an MLR reaction. One needs to know
the mechanism of poor MLR response.

*We agree with the Reviewer that an impaired mixed lymphocyte reaction (MLR) should*
*correlate with a decreased expression of HLA-DR, as previously reported [12].*

*We showed that CD14⁺ DCs always display lower HLA-DR surface levels compared to cDC2s*
*(Fig. 4d), and that HLA-DR expression progressively downregulates over time (8h vs 48h)*
*(Suppl. Fig 7b,c). In this revised manuscript we included relative percentages of CD3⁺ T cell*
*proliferation (Fig 6d), and a direct correlation between MLR stimulatory capacity and HLA-DR*

expression (Fig. 6e). Together, this data confirms that the degree of HLA-DR expression in both
 CD14⁺ DCs and cDC2s is directly associated with their ability to stimulate lymphocyte
 proliferation. Text (page 10) and Figure legend (pages 28-29) have been modified accordingly.

Minor points:

1. In Fig 5, authors have used CD25 as an activation marker. One should also use HLA DR and
 CD69 markers.

As suggested by the reviewer, CD69 and HLA-DR antigens are, together with the alpha chain
 of the IL-2 receptor (CD25), important T cell activation markers. However, these markers have
 different kinetics of up-regulation upon T cell activation.

In the original manuscript (Fig. 6f and text on page 10), we showed that the expression of HLA-
 DR and CD25 was lower on CD3⁺ T cells stimulated by autologous CD14⁺ DCs, compared to
 cDC2 DCs. Within the same experiments, we also checked the levels of expression of CD69 on
 5 day-cultured CD3⁺ T cells. Although we measured statistically significant differences, similar
 to those observed for HLA-DR and CD25, the frequency of CD69 positive cells was too low to
 draw any meaningful conclusion (please see rebuttal Fig. 2a). Similarly, analysis of CD3⁺ T cells
 co-cultured with either cDC2s or CD14⁺ DCs overnight (approximately 16h) showed only a
 slight increase in the percentage of CD69⁺ CD3⁺ T cells (please see rebuttal Fig. 2b).

Given the low cell frequencies detected, we believe that these results are less important to the
 main message of the manuscript. Therefore, data are only shown below for the Reviewer's
 convenience.

**Rebuttal Figure 2.**

**a**, Autologous T cells were co-cultured with freshly isolated and tumour-conditioned cDC2s and CD14⁺
 DCs for 5 days and then stained for CD69, CD3, CD8 and live/dead marker. Percentage of activated
 CD69⁺ cells in total CD3⁺ T cells is reported. **b**, Autologous T cells were co-cultured overnight with freshly

*isolated and tumour-conditioned cDC2s and CD14⁺DCs and then stained for CD69, CD3, CD8 and*
*live/dead marker. Percentage of activated CD69⁺ cells in total CD3⁺ T cells is reported. Box and whiskers*
*graph (n. of replicates: at least 2, min to max, show all points) shows the result of two independent*
*experiments. Each symbol represents an individual donor. (mean±SEM; one-way ANOVA and Tukey's*
*post hoc correction). Statistical significance was annotated as follows: **p < 0.01, ***p < 0.001, ****p*
*< 0.0001.*

2. In Fig 1, authors need to provide scale bars instead of magnifications.

*Scale bars have been included.*

a. In the same Fig, the authors need to provide arrows to indicate different cell types in the
organotypic cultures.

*Arrows have been included to discriminate fibroblasts from melanoma cells within the OMC,*
*as well as in human melanoma lesions stained for comparison.*

3. In Fig 3, how many DCs interaction with tumor cells during a given time?

*To address the Reviewer's question, we quantified the percentage of DCs (magenta)*
*interacting with tumour cells (BLM-GFP, green) in OMCs, over 30-min time-lapse 2-photon*
*microscopy videos. The image processing- and quantification- steps are depicted in Suppl. Fig*
*5d-f. Five fields-of-view (FOVs) from 2 independent experiments were analysed, with a time-*
*lapse interval of 2min. In each FOV, the area occupied by tumour cells represents 10% to 20%*
*of the total imaging area (Suppl. Fig 5e). Maximum intensity projections of 3 consecutive z-*
*stacks, for a total z-size of 10µm were generated. The semi-automated quantification of DC-*
*tumour interactions is based on the measurement of GFP pixel intensities (tumour) that*
*overlay with DC selections. As shown by the quantification, DC interaction with tumour cells is*
*a highly dynamic process, and the percentage of DCs that interacted with tumour cells during*
*a given time is comprised between 20% and 60% of the total DCs in the FOV (Suppl. Fig 5f).*

*M&M (pages 21-22) and Supplementary figure legend (pages 6-7 of Supplementary material)*
*have been modified accordingly.*

**Reviewer #2 (Remarks to the Author):**

In this manuscript Di Balsio and co-authors described a novel human organotypic skin
melanoma culture (OMC). They co-cultured decellularized dermis with keratinocytes,
fibroblasts and immune cells in the presence of melanoma cells. Thus, they tried to generate
a reconstructed tumor microenvironment. They demonstrated that the OMC is suitable and
outperforms conventional 2D co-cultures for the study of dendritic cells dysfunction.

The presented experimental model is novel and potentially useful. Experiments are clearly
described and conclusion are convincing. The potential usage of this system for the analysis
of immune mechanisms is not clear at this moment since presented data are rather limited.

However, publication of these results would open the possibility for scientific community to
test it and thus would be desirable.

*We thank the reviewer for his/her positive remarks, and for acknowledging the novelty and*
*relevance of the organotypic model we developed to study immunomodulation in the context*
*of skin cancer. We agree that the mechanism of immune cell conversion was not resolved. In*
*the revised manuscript we have now included data that indicate which soluble factors,*
*produced within the OMC, contribute to the induction of the immunosuppressive CD14⁺ DCs.*
*Moreover, we have now demonstrated that the model is also suitable for culture and*
*characterization of fresh tumour specimens (please see below reply to Reviewer 3-Question*
*1), and to investigate changes in autologous immune cells infiltrating the TME. We hope to*
*have provided evidence of the potential of the OMC for its use by a broad scientific community.*

**Reviewer #3 (Remarks to the Author):**

In the study “The tumour microenvironment shapes dendritic cell plasticity in a human
organotypic melanoma culture”, Figdor and colleagues describe an in vitro model to study
how the tumour microenvironment influences the immune system. This is interesting work
with potential clinical significance. However, there are several issues that limit my enthusiasm
and need to be addressed.

Specific comments:

1. Only melanoma cell lines are used in the OMCs, which still makes it a very artificial model.
It would be much more valuable if primary patient-derived melanoma cells as well as immune
cells from the same patient can be implemented in the model.

*We agree with the Reviewer that the use of patient-derived tumour and autologous immune*
*cells would be absolutely relevant and required. In this respect, in the original manuscript we*
*showed that melanoma tumour suspension can grow within the dermal scaffold; and that*
*tumour-infiltrating lymphocytes remained not only viable over time, but also maintained*
*frequencies similar to that observed prior to injection (Fig. 8a-b, Suppl. Fig. 12a).*

*In the revised manuscript, we have extended these observations with additional freshly*
*isolated tumour cells derived from resected material of cancer patients. Subsequently, OMC*
*were formed and studied in time. We assessed expression of HLA class-I, different melanoma-*
*associated antigens (MART-1/MELAN-A, Gp100/HMB45, Tyrosinase), SOX10 and Ki67 to*
*characterize melanoma cells in the surgically removed tumour material used in this study. The*
*newly included Figures 8c-d (and Suppl. Fig 13) show the comparative analysis performed on*
*the original melanoma lesion and on patient-reconstituted OMC. Altogether, these data*
*support the data obtained with the melanoma cell lines and the validity of our model.*

*Furthermore, we assessed phenotypic modulation of myeloid cells in an autologous setting as*
*requested by the reviewer, using melanoma patient material (Fig. 8e, Suppl. Fig. 12b). We*
*prospectively collected fresh tumour specimens and blood from melanoma patients (n=3), to*
*obtain pairs of autologous tumour suspensions and peripheral blood derived immune cells.*

*The CD45⁻ tumour fraction, containing both tumour and stromal cells, was allowed to grow*
*within the OMC. Next, CD14⁺ monocytes were injected into the OMC. In these experiments we*
*used monocytes, instead of cDC2 DCs, as a source of myeloid cells, because of the very limited*
*volume of blood obtained. This did not allow us to isolate sufficient numbers of cDC2, which*
*comprise only 0.6% to 1% of the mononuclear cell fraction. We hope that the reviewer will*
*appreciate this, given the ethical limitations to obtain large amounts of blood from patients.*
*In line with the described tumour-driven monocyte modulation [14], we observed down-*
*regulation of HLA-DR, and the concomitant up-regulation of the CD14 and MerTk*
*monocytic/macrophage markers and PD-L1 on CD14⁺ cells (Fig. 8e). This is consistent with our*
*observations that the tumour environment within the OMC drives conversion of cDC2 by down*
*regulation of HLA-DR and upregulations of CD14.*

*Text (pages 11-12), Figure legend (page 29) and Supplementary Figure legend (page 9 of*
*Supplementary material) have been modified accordingly.*

2. Figure 1 describes OMC establishment. Yet, the OMCs are not well characterized. Are the
melanoma cells still proliferating in the OMCs and how does this compare to primary tumour
tissue? The images in Figure 1C are not clear and primary tumour lesion should be included
as comparison. Lastly, scale bars are missing.

*We thank the Reviewer for these suggestions. In agreement with Reviewer #1's request, Fig.*
*1c now includes a better IHC staining of fibroblasts and tumour cells within the OMC, and a*
*comparative staining on human melanoma tissue. We modified Fig. 1b and included a triple*
*Ki67/tumour marker/DAPI staining on both melanoma lesion and OMC sections. Alongside, to*
*document the proliferation of melanoma cells growing in the OMC we conducted flow*
*cytometry intracellular Ki67 staining on the CD45⁻ fraction, of the digested OMC suspension.*
*A representative histogram reporting the percentage of Ki67⁺ cells within the CD45⁻ population*
*is now part of Fig. 1b. Text (page 6) and Figure legend (page 26) have been modified*
*accordingly.*

3. To authors claim that DCs cells are migrating in the OMC by showing their distribution
throughout the tissue sections 2 days after injection. However, to make such a claim the
authors at least need to show DC distribution right after injection as well. Are they randomly
migrating or moving towards tumour cells?

*We agree with the Reviewer that the distribution of DCs right after injection was not shown.*
*To address this question, we have injected DCs in OMCs, following standard protocol: 3-*
*microneedle arrays were used to perform 3 injections, onto the dermal side of the OMC, in*
*different locations (Rebuttal Fig. 3a, Suppl. Video 1). Thus, OMCs were fixed either*
*immediately after injection ("0h") or 2days post injection ("48h") (n=2). OMCs were*
*subsequently sectioned and immunostained for identification of CD45⁺ DCs. On CD45 stained*
*sections, positive cells were annotated over at least 5 layers (=depths), and DC distribution*
*within the section (X-Y distribution) was assessed by Ripley's K clustering coefficient. A higher*
*Ripley's K coefficient indicates more cell clustering. As shown by the quantification of DC*
*clustering at 0h, performing multiple injections ensure a good X-Y distribution of DCs already*

after injection (Rebuttal Fig 3b). DC distribution (X-Y distribution) does not significantly change
over time (48h, Rebuttal Fig 3b).

We also agree that it would be important to define the migratory nature of DCs towards
tumour cells. However, in order to understand whether DCs respond to chemotactic signals or
they randomly migrate throughout the tissue, the use of culture models specifically designed
for chemotaxis-related migratory response would be better suited. In particular, within the
OMC, tumour and stromal cells are found in multiple regions of the tissue, as a result of their
random distribution and growth upon cell seeding (please refer to Fig 2b of the manuscript).
Moreover, the current manner of injection of DCs using a microarray of three needles, to
perform three injections at different locations, ensures DC dispersion but makes it impossible
to control their tissue distribution (Rebuttal Fig 3b, manuscript Fig 2b). On the contrary, studies

**Rebuttal Fig 3.**

**a**, Schematic representation of cell distribution within OMCs, prior to DC injection and after DC
injection. DCs are injected using a 3-needle microarray system, 3 consecutive injections are performed
in different locations, from the dermal side of the OMC. Due to random tumour cell dispersion upon
cell seeding, a multidirectional chemotactic gradient is likely to generate within the OMC environment.
For technical details about injection, please also refer to Suppl. Video 1. **b**, Analysis of DC clustering in
CD45-stained tissue sections immediately after injection (0h) and 2 days after injection (48h). Ripley's
K clustering coefficient was calculated for each tissue section, representing different depths within the
tissue, in two independent experiments.

that aim to address cell migration towards a chemotactic stimulus in a three-dimensional
tissue matrix, require that location of both tumour cells (source of the chemotaxis gradient)
and DCs (responders) is controlled and restricted to opposite areas of the matrix. For those
reasons, although it might provide a suitable platform for chemotactic studies, the OMC
culture as described in this manuscript would need to be optimized for the specific

*experimental question. Albeit feasible, we hope that the Reviewer might understand that this*
*would be a whole new study. Two-photon microscopy imaging revealed that DCs are motile*
*with a typical hand mirror shape within the reconstructed TME, where they also interact with*
*tumour cells (please see replies to Reviewer 1-Question 3 and Reviewer 3-Question 4).*

*Text in the manuscript has been modified to highlight this phenomenon, but exclude any*
*reference to DC migration throughout the reconstructed tissue (page 7).*

*4. In line with my previous comment, it is unclear from Figure 3 if the interaction of DCs with*
*melanoma cells and particles derived thereof is just a stochastic process, or whether the DCs*
*are actively moving towards them. What percentage of DCs show a migratory “hand mirror*
*shape” phenotype and are these typically the ones that are in close proximity of tumour cells?*

*We thank the Reviewer for this suggestion. As discussed in the previous comment, the 3D*
*multicellular complexity of the OMC, and in particular the technical difficulty in controlling the*
*distribution of tumour cells in the reconstructed TME, does not allow us to determine whether*
*the interaction of DCs with tumour cells is stochastic or rather driven by a chemotactic*
*gradient, secreted by a tumour.*

*To address the Reviewer’s question on cellular shape, we have analysed DC morphology*
*(magenta), with respect to their proximity to a tumour (BLM-GFP, green), over 50-min time-*
*lapse 2-photon microscopy videos. The image processing and quantification steps are depicted*
*in Suppl. Fig 5a-c and described in the M&M section (pages 21-22).*

*In brief, five fields-of-view (FOVs) from 2 independent experiments were analysed. In each*
*FOV, the tumour area represented 10% to 20% of the total imaging area. Maximum intensity*
*projections of 3 consecutive z-stacks, for a total z-size of 10µm were generated. Regions of*
*interest (ROIs) were drawn to define DCs in proximity to tumour (distance <100µm,*
*“Proximal”) and DCs distant from tumour (distance >100µm, “Distal”). A semi-automated*
*analysis of cell shape parameters within either Proximal or Distant ROIs was performed.*
*“Roundness” is a measurement of how cell shape is close to a circle (whereby 1= circular cell,*
*0= elongated cell), and it has been used to indicate degree of cell protrusion. In our analysis,*
*a roundness factor of approximately 0.6 was calculated in cells with a typical “hand mirror*
*shape” phenotype (Suppl. Fig 5b). We therefore analysed DC morphology in time-lapse images*
*with a time-lapse interval of 10min. Results from two experiments are shown in Suppl. Fig 5c.*
*Albeit not significant, proximal DCs have lower roundness values compared to distal DCs (from*
*30 min after start of imaging acquisition).*

*The cellular microenvironment, such as the extracellular matrix structure and the presence of*
*a chemotactic stimulus, as well as the specific properties of a cell cytoskeleton, are key*
*determinants of cellular locomotion and adaptation to dynamic conditions [14]. We agree*
*with the Reviewer, that understanding whether the chemotactic gradient generated by a*
*tumour, reflects changes in migratory immune cell behaviour and ameboid phenotype would*
*certainly be an interesting new study.*

*Text (pages 7-8) and Supplementary figure legend (pages 6-7 of Supplementary material) have*
*been included accordingly.*

5. The authors present the OMC model as a platform for exploring cancer-driven immune cell
modulation. In Supp Fig 8, they demonstrate that human lymphocytes can be maintained in
the OMCs as well. The manuscript would greatly benefit from some additional experiments
demonstrating the applicability of OMCs to study immune modulation, such as DC-dependent
cytotoxic T-cell stimulation and the effect of checkpoint inhibition on tumour cell killing.

*Data that were presented in Suppl. Fig. 8 of the original manuscript have been now extended,*
*demonstrating that the OMC model is also very valuable to analyse the immunosuppressive*
*capacity of a patient's tumour microenvironment. We now also show how peripheral blood*
*monocytes undergo the same phenotypic conversion as we observe for cDC2 DCs in OMCs.*
*Moreover, we showed that primary tumour cells grown in the model retain immunogenicity*
*and proliferation capacity, certainly supporting the applicability of the OMC for T cell-*
*mediated antigen recognition and killing. Albeit feasible and certainly interesting, we hope*
*that the Reviewer agrees with us that such experiments would require a dedicated new setup*
*of assays and conditions such as: usage of checkpoint inhibitory drugs within the 3D system,*
*dedicated 2-photon experiments and IHC analysis for the evaluation of T cell-DC interaction,*
*and the assessment of the favourable time-points for the analysis. We certainly believe this is*
*an interesting new study.*

**References**

- 1. Eves, P. et al. Melanoma invasion in reconstructed human skin is influenced by skin cells--
investigation of the role of proteolytic enzymes. *Clin. Exp. Metastasis* 20, 685-700 (2003).
- 2. Gibot, L., Galbraith, T., Huot, J. & Auger, F. A. Development of a tridimensional
microvascularized human skin substitute to study melanoma biology. *Clin. Exp. Metastasis* 30,
83-90 (2013).
- 3. Li, L., Fukunaga-Kalabis, M. & Herlyn, M. The three-dimensional human skin reconstruct
model: a tool to study normal skin and melanoma progression. *J. Vis. Exp.*(54). pii: 2937. doi,
10.3791/2937 (2011).
- 4. Meier, F. et al. Human melanoma progression in skin reconstructs: biological significance of
bFGF. *Am. J. Pathol.* 156, 193-200 (2000).
- 5. Syed, D. N. et al. Fisetin inhibits human melanoma cell growth through direct binding to
p70S6K and mTOR: findings from 3-D melanoma skin equivalents and computational
modeling. *Biochem. Pharmacol.* 89, 349-360 (2014).
- 6. Bartoschek, M. et al. Spatially and functionally distinct subclasses of breast cancer-associated
fibroblasts revealed by single cell RNA sequencing. *Nat. Commun.* 9, 5150-018-07582-3 (2018).
- 7. Strutz, F. et al. Identification and characterization of a fibroblast marker: FSP1. *J. Cell Biol.* 130,
393-405 (1995).
- 8. Nurmik, M., Ullmann, P., Rodriguez, F., Haan, S. & Letellier, E. In search of definitions: Cancer-
associated fibroblasts and their markers. *Int. J. Cancer* 146, 895-905 (2020).
- 9. Huber, V. et al. Tumor-derived microRNAs induce myeloid suppressor cells and predict
immunotherapy resistance in melanoma. *J. Clin. Invest.* 128, 5505-5516 (2018).
- 10. Mao, Y. et al. Melanoma-educated CD14+ cells acquire a myeloid-derived suppressor cell
phenotype through COX-2-dependent mechanisms. *Cancer Res.* 73, 3877-3887 (2013).
- 11. Gabrilovich, D. I., Ostrand-Rosenberg, S. & Bronte, V. Coordinated regulation of myeloid cells
by tumours. *Nat. Rev. Immunol.* 12, 253-268 (2012).

- 12. Shaked, A., Hoyos, B. & Mayer, L. *The role of differential class II antigen expression in*
*stimulation of allogeneic mixed lymphocyte reactions by human monocyte hybridomas.*
*Transplantation* 53, 1341-1347 (1992).
- 13. Ge, Q., Palliser, D., Eisen, H. N. & Chen, J. *Homeostatic T cell proliferation in a T cell-dendritic*
*cell coculture system.* *Proc. Natl. Acad. Sci. U. S. A.* 99, 2983-2988 (2002).
- 14. Yoshida K. and Soldati T. *Dissection of amoeboid movement into two mechanically distinct*
*modes.* *J Cell Science.* 119, 3833-3844 (2006).

REVIEWERS' COMMENTS:

Reviewer #1 (Remarks to the Author):

This revised manuscript has considerably improved and authors have addressed all the comments raised by the reviewers.

Additional experiments added to the manuscript confirms authors' original claims of conversion of DC subtype.

Reviewer #2 (Remarks to the Author):

Authors addressed my comments satisfactory.

Reviewer #3 (Remarks to the Author):

The authors have done a significant number of additional experiments and have addressed the majority of my concerns. However, the technical difficulties that are put forward by the authors, not allowing them to fully address my comments, are currently underexposed in the manuscript. Pros and cons of the model should be discussed more elaborately.

Reply to Reviewer #3' comment:

Reviewer #3 (Remarks to the Author):

The authors have done a significant number of additional experiments and have addressed the majority of my concerns. However, the technical difficulties that are put forward by the authors, not allowing them to fully address my comments, are currently underexposed in the manuscript. Pros and cons of the model should be discussed more elaborately.

With the aim to further elaborate on pro's and con's of the OMC culture as described in this manuscript, we included new paragraphs in the Discussion session (page 14, lines 455 and 466).

We emphasized the advantage of the model, as a complex 3D tissue culture system, in recapitulating clinical features of primary tumour specimens, and facilitating the investigation of the effects of a tumour microenvironment (composed of more than 2 cell types, in an extracellular matrix) on immune cell functionality. Those two aspects would not be reproducible in 2D conventional cultures.

On the other hand, we are well aware that, like most organotypic culture systems, the OMC model presents some limitations. Such limitations could be overcome by further optimization. In particular, it would be very interesting to apply the OMC to address chemotaxis-related migratory response of DCs towards tumour cells. In the current settings, where tumour cells are dispersed throughout the dermal scaffold as a result of the seeding approach, it would be difficult to dissect the contribution of a tumour-derived chemotactic gradient to the migratory behavior of DCs.

In Suppl. Fig 5, we showed that DCs are highly dynamic and adapt to their surrounding environment by adopting different cell shapes, as they move along and through the collagen fibers. Thus, the OMC can potentially offer a platform for the advanced investigation of migratory events in a local skin environment.